# Integrative genomic analyses of promoter G-quadruplexes reveal their selective constraint and association with gene activation

Guangyue Li [1], Gongbo Su [1], Yunxuan Wang[2], Wenmeng Wang[1], Jinming Shi[1], Dangdang Li [1] & Guangchao Sui [1✉]

G-quadruplexes (G4s) regulate DNA replication and gene transcription, and are enriched in promoters without fully appreciated functional relevance. Here we show high selection pressure on putative G4 (pG4) forming sequences in promoters through investigating genetic and genomic data. Analyses of 76,156 whole-genome sequences reveal that G-tracts and connecting loops in promoter pG4s display lower or higher allele frequencies, respectively, than pG4-flanking regions, and central guanines (Gs) in G-tracts show higher selection pressure than other Gs. Additionally, pG4-promoters produce over 72.4% of transcripts, and promoter G4-containing genes are expressed at relatively high levels. Most genes repressed by TMPyP4, a G4-ligand, regulate epigenetic processes, and promoter G4s are enriched with gene activation histone marks, chromatin remodeler and transcription factor binding sites. Consistently, *cis*-expression quantitative trait loci (*cis*-eQTLs) are enriched in promoter pG4s and their G-tracts. Overall, our study demonstrates selective constraint of promoter G4s and reinforces their stimulative role in gene expression.

[1] College of Life Science, Northeast Forestry University, Harbin 150040, China. [2] Department of Medical Oncology, Harbin Medical University Cancer Hospital, Harbin 150081, China. ✉email: gcsui@nefu.edu.cn

Genome-wide association studies (GWAS) have revealed that over 88% of the genetic variations associated with different diseases are located in the noncoding regions of the human genome[1]. The regulatory elements affected by variations mainly include promoters, enhancers and transcription factor binding sites that are involved in transcription machinery assembly and chromatin remodeling, eventually altering gene expression.

A promoter is generally considered as a DNA sequence with a certain length upstream of the transcription start sites (TSSs) of a gene and is a functional element associated with various regulatory factors. Gene expression can be either activated or inhibited by the presence of transcription factor binding sites, DNA methylation and histone modification statuses in promoters[2]. In addition, levels of non-B-DNA structures, such as G-quadruplexes (G4s), are altered in cancer cells and tissues compared to normal counterparts, and the formation and stabilization of these special DNA structures can determine transcription levels of target genes, which provides insights for therapeutic interference of cancer development[3,4].

A G4 motif is composed of at least four G-tracts, each of which includes three or more guanines (Gs), and three loops linking adjacent G-tracts typically with 1-12 nucleotides (nts) in each, leading to the consensus motif sequence of $G_{3+}N_{1-12}G_{3+}N_{1-12}G_{3+}N_{1-12}G_{3+}$[5]. The four consecutive G-tracts are the most essential elements for G4 structure formation, and self-recognized Gs from these G-tracts can form stacked G-tetrads that are stabilized by Hoogsteen bonds and central cations. G4 stability is determined by many factors, including G-tract length, G-tetrads, loop length and structure, and composition of the sequences within and flanking G4s[6,7]. Due to the highly stable structures, G4s can impede the progression of DNA replication by promoting replication-fork collapse[8], leading to DNA double-strand breaks and increased genomic variations. Meanwhile, G4 structures also obstruct the processes of RNA transcription and protein translation[9]. RNA G4s are important regulators of pre-mRNA processing, including splicing and polyadenylation[10]. G4 motifs are frequently identified in exons, especially in regions close to splicing sites; consistently, splicing quantitative trait loci (sQTLs) are enriched in G4 sequences, which is evolutionarily conserved across many species[11].

The impediment of G4 structures to various chromatin-related processes explains their potence in impairing genome stability. Putative G4s (pG4s) are enriched at the loci of germline and somatic mutations, and frequently cause DNA sequencing errors[12,13]. Helicases, such as PIF1 and BLM that unfold G4 structures, can improve genome stability[14,15], while other non-helicase G4-interacting proteins may either prevent or augment G4-associated genome disturbance[16]. The substitutions of any G in a G-tract, especially the central one(s), by another nucleotide can cause detrimental effects on G4 stability and are associated with pathological consequences, such as the alterations in untranslated regions (UTRs)[17] and telomeres[18]. Overall, G4s are causally associated with genome instability, and G4 loci in genome generally have high variation density and disruptive sites with low minor allele gene frequencies (MAFs), supporting their evolution under negative selection[19].

Previous studies suggested that pG4s represent a type of drivers in genome evolution (reviewed in ref. [20]). G4 loci exhibited heightened selective pressure and the middle Gs of 3G G-tracts in UTRs has a similar degree of selective pressure to missense variations in protein-coding regions[17]. Another study reported the selective pressure of pG4s at different genic positions, including promoters; however, the contributions of G-tracts or loops, which are essential elements for G4 folding, have not been individually examined[21]. Due to distinct sequence features, it is still possible that G-tracts and loops are subject to different selective pressures.

Through massively parallel reporter assays (MPRAs), G4s were shown to enhance promoter activity, but for unknown reasons the 2.5 kb regions flanking TSSs were excluded from the analyses[22,23]. Experimental studies indicated that many promoter G4s could either activate or repress downstream gene expression[9]. However, pG4's effects on promoter activity in a large-scale and systematic study have not been reported.

Based on previous studies, we asked whether different parts of a G4 in promoters are subject to distinct selective pressures, which may affect chromatin state and determine promoter activity. To answer these questions, we combinatorially analyzed multiple large-scale genomic and genetic datasets from different databases to evaluate the evolutionary constraint on promoter pG4 sequences in the human genome, and enrichment of functional relevance. We discovered that promoter pG4 sequences are under negative selection and G-tracts in pG4 sequences are exposed to heightened selective pressures compared with whole pG4s. Additionally, promoter G-tracts are enriched with *cis*-eQTL variants, histone modifications, and transcript factor (TF) and chromatin remodeler binding sites. Together, our results revealed the biological relevance of promoter pG4 sequences, especially their interior G-tracts, and reinforced the critical roles of chromatin secondary structures in the regulation of gene expression.

## Results

**Characterization of G4s in promoters**. A G4 structure consists of at least four G-tracts and three connecting loops (Fig. 1a). In the genome, a consensus G4 motif is a prerequisite for G4 formation but may not always form a G4 structure, depending on its sequence context and microenvironment. Therefore, we designated a DNA sequence consisting of a G4 motif as a putative G4, or pG4. Using the Quadron software[7] to analyze the human reference genome hg38, we discovered a total of 707,816 pG4s, of which about 97.9% had their lengths fall between 15-76 base pairs (bps). Each pG4 sequence contains 4-8 G-tracts, 410,844 G4s include only 4 G-tracts, and the median stability score was 19.47 based on Quadron analysis (Supplementary Fig. 1). We defined the G4s with predicted scores of >19 as stable-pG4, and those with scores ≤19 as unstable-pG4. Additionally, when analyzing G4s identified by DNA sequencing based on DNA synthesis retardation caused by G4 structures[24], we found that 81.42% of them were stable-pG4s based on the Quadron prediction (Supplementary Fig. 2).

A promoter is the DNA region upstream of the transcription start site (TSS) of a gene, and can bind various regulatory proteins, such as transcription factors (TFs) and RNA polymerase, to initiate RNA synthesis. The length of a promoter is typically 100–1000 bps, and can be extended based on the abundancy of TF binding sites. We designated the (−1000 to 0) region of each gene as its "1 kb" (1 kilobase pairs) promoter, when the first nucleotide upstream of the TSS was designated as −1. Meanwhile, we also defined the 200 bps upstream of each TSS as its "0.2 kb" promoter, or active promoter (Fig. 1b), based on the promoter definition from the ENCODE using the epigenetic marks of H3K4me3 and H3K27ac modifications, CTCF binding sites, and DNase sensitive regions[25].

One gene may have several promoters due to different transcription initiation sites of its multiple transcripts. In 19,955 protein-coding genes from GENCODE database, we totally identified 41,259 promoters, because 10,999 of these genes contain two or more promoters. Next, in the 17,343 promoters of 13,657 genes, we discovered 38,680 pG4 sequences. The

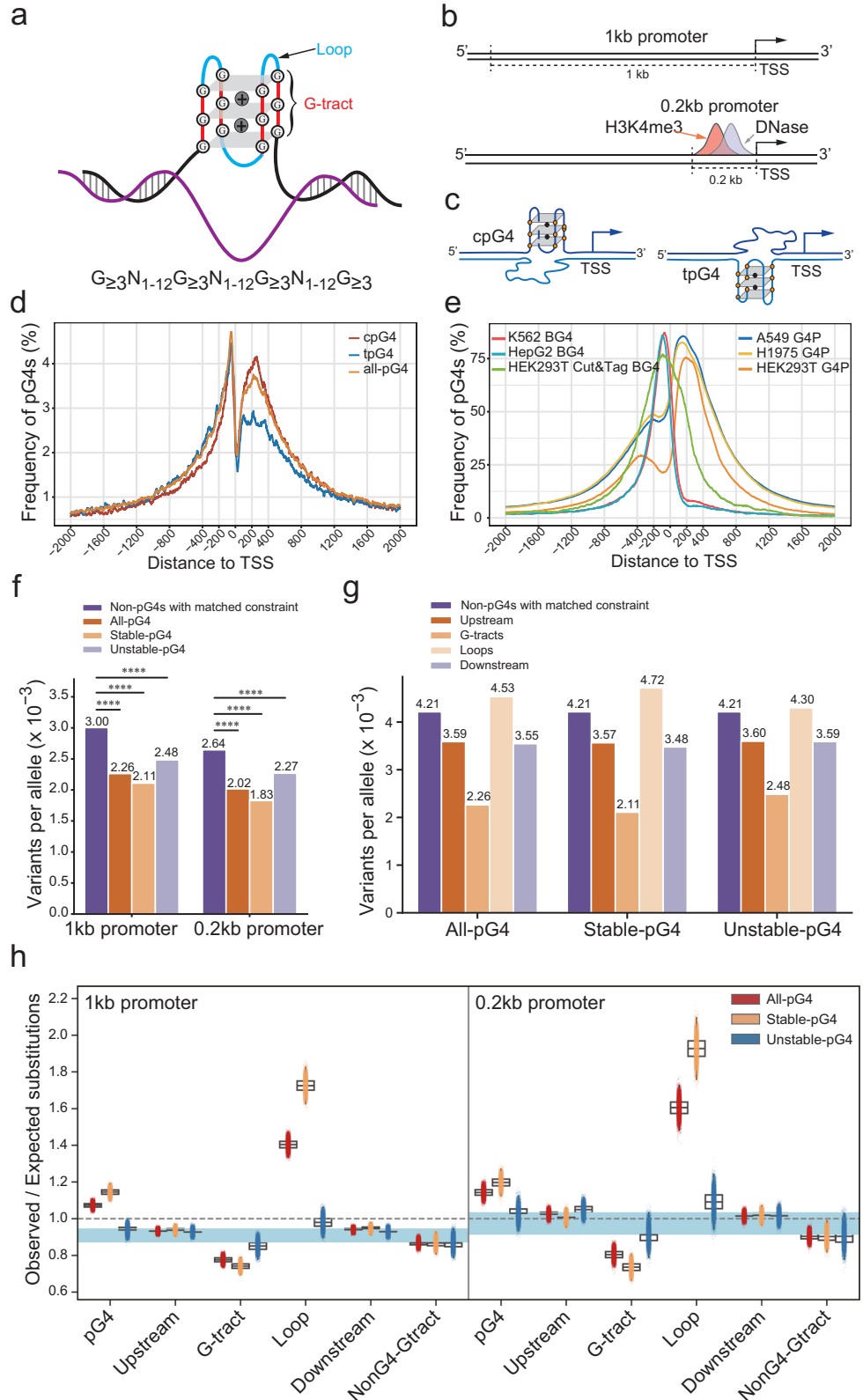

average length of pG4 motifs in promoters ranges from 15 to 91 bps, and the median value of the thermostability score was 21.94 (Supplementary Fig. 3a, b).

A total of 21,964 0.2 kb promoters were discovered in 16,119 protein-coding genes, and 5126 genes contained two or more 0.2 kb promoters. Among them, 10,730 0.2 kb promoters in 9,510 genes harbored 16,035 pG4 sequences. The length distribution of

pG4s in 0.2 kb promoters was from 16 to 80 bps, and the median thermostability score was 22.15 (Supplementary Fig. 3c, d).

Previous studies indicated that both pG4s and experimentally identified G4s are enriched in the promoter regions of the human genome[26,27]. However, detailed distribution of G4s in the proximity of TSSs and their structural stability have not been characterized. The presence of pG4s in either the coding or

**Fig. 1 G4 schematic structures, positions, distribution within 2 kb flanking regions of TSSs and pG4's allele frequencies in promoters. a** Schematic depiction of a folded G-quadruplex with 3G G-tracts and a formula of G4 motifs. Each G-tract contains at least three consecutive Gs and a loop contains the sequences between two adjacent G-tracts. **b** Schematic diagrams of the 0.2 kb and 1 kb promoters. The 0.2 kb promoter and 1 kb promoter are defined as the regions of 200-bp and 1-kb upstream of the TSS of a transcript, respectively. Transcription start sites (TSSs) and transcription directions are labeled, and the distribution of H3K4me3 mark and DNase sensitive sites in the 0.2 kb promoter are shown. **c** Schematic representation of the positions of cpG4s and tpG4s. pG4s formed by the coding strand and template strand of a gene are defined as tpG4s and cpG4s, respectively. **d** Distribution of tpG4s, cpG4s and both within the 2-kb regions upstream and downstream of TSSs. **e** Distribution of G4s identified by BG4 antibody or a G4 probe (G4P, an artificial G4 binding protein) using different techniques within 2-kb upstream and downstream of TSSs. **f** Comparison of variant frequencies affecting G-tracts in different promoter pG4 forming sequences compared to matched non-pG4 G-tracts with transcript-level constraint. Stable-G-tracts and unstable-G-tracts are those within promoter pG4s with thermostability scores higher or lower than 19, respectively ($P < 2.2 \times 10^{-16}$ by Chi-Squared test). **g** Allele frequencies in G-tracts, loops, 100 bp regions upstream and downstream of pG4s, and LOEUF-constrained non-PG4 regions. **h** Ratios of observed versus expected polymorphic sites in different pG4s, their partitioned sections and upstream/downstream regions in 1 kb and 0.2 kb pG4-containing promoters using a nucleotide substitution model based on local sequence context. Each box plot (central line, median, box limits, upper and lower quartiles) represents the ratio of observed versus expected substitutions within each pG4 region. The light blue shaded regions represent the observed versus expected numbers of substitutions in non-pG4 promoter sequences matched by transcript-level constraint with 95% confidence intervals. The dashed line depicts the 1:1 ratio of expected versus observed polymorphic sites.

template strand, designated as coding strand pG4s (cpG4s) and template strand pG4s (tpG4s), respectively (Fig. 1c), may have different biological relevance. Based on the consensus G4 motifs, we calculated G4 frequencies in the TSS-centered 4 kb-regions in the genome and discovered that pG4s were enriched within 1 kb regions flanking the TSSs. The distribution of cpG4s and tpG4s showed similar patterns upstream of TSSs, although tpG4s exhibited slightly higher frequency than cpG4s in the region from −2 kb to −100 bps (Fig. 1d). Downstream of TSSs, cpG4s showed markedly higher frequency than tpG4s (Fig. 1d), suggesting that G4s are enriched in the 5'-untranslated regions (5'-UTRs) and/or the first exons of genes, which are key regulatory regions of gene expression at the posttranscriptional level. Consistently Lee, et al. also reported G4 enrichment in 5'-UTR regions[17].

We further analyzed several publicly available ChIP-seq datasets that were obtained using a single-chain antibody recognizing G4 structures (BG4), a G4 probe (G4P, an artificial G4 binding protein), or BG4-associated Cut & Tag method (Cut&Tag-BG4). All these methods indicated that G4s are highly enriched in the vicinity of TSSs despite their relatively different peak positions. In three BG4-related studies, two datasets obtained by BG4 in K562 and HepG2 cells[28,29] and one dataset collected by the Cut&Tag-BG4 method in HEK293T cells[30] showed peaks around −100 bps, although the curves differed in their breadths (Fig. 1e). However, based on the three ChIP-seq datasets using the G4P in A549, H1975 and HEK293T cells[31], G4s showed bimodal distributions around the TSSs with the major peaks between 0 and 200 bps and secondary peaks between −400 and −200 (Fig. 1e).

**pG4s exhibit heightened selective pressure within promoters.** Assuming that pG4s are biologically functional, they should exhibit patterns of high selection constraint in evolutionary terms. To test this hypothesis, we evaluated the single nucleotide variants (SNVs) within promoter pG4 motifs using the whole genome sequencing data of 76,156 individuals from the public gnomAD release (version 3.1.2)[32]. If promoter pG4s experience heightened selective pressure, we expect that the variants with altered promoter pG4 sequences will show relatively low allele frequency compared to those of the variants with non-pG4 promoters.

G-tracts with three or more consecutive Gs are essential for G4 formation. We compared the SNVs affecting the G-tracts of pG4s to those in the G-tracts of non-pG4 (≥3 G) within promoters that belong to the transcript subset with estimated overall constraint levels matching our promoter containing transcripts. This subset of reference transcripts with non-pG4 G-tracts was selected using

the upper 90% bound of the observed versus expected (LOEUF) metric, as published by gnomAD[32] (Supplementary Fig. 4). As shown in Fig. 1f, compared to the variants in G-tracts of non-pG4s from constraint-matched transcripts, the variants in the G-tracts of all-pG4s in 1 kb promoters and 0.2 kb promoters showed significant allele frequency reduction (from 3.00 to 2.26, and from 2.64 to 2.02, respectively). For these 1 kb and 0.2 kb promoters, more decreases (to 2.11 and 1.83, respectively) in the G-tracts of stable-pG4s, but less (to 2.48 and 2.27, respectively) in the G-tracts of unstable-pG4s were observed (Fig. 1f). Based on Chi-Squared test, all these allele frequency differences in G-tracts were highly statistically significant ($P < 2.2 \times 10^{-16}$). Overall, the reduced allele frequencies of G-tract variants in these pG4s of 1 kb and 0.2 kb promoters relative to those in non-pG4s imply the effects of negative selection.

Allele frequencies across the genome can be affected by local sequence contexts and constrained by functional elements of nearby linkage selection. Therefore, we partitioned each pG4 in 1 kb promoters into G-tracts, loops and bilateral regions (including upstream and downstream of pG4), to analyze their individual allele frequencies. In all-, stable- and unstable-pG4s, G-tracts exhibited the lowest allele frequency, and loops showed the highest, while the allele frequencies of the regions upstream and downstream of pG4s were lower than those of the sequences in non-pG4 promoters (Fig. 1g). Similar results were obtained in the analysis of partitioned pG4 sections in 0.2 kb promoters (Supplementary Fig. 5).

Next, to understand the selective pressure of pG4s in promoters, we applied a background model of neutral evolution to evaluate SNP distributions in 1 kb and 0.2 kb promoters under the neutral theory of molecular evolution, which was used to explain >81% of variability in substitution probabilities for noncoding genomic regions based on the local hepta-nucleotide context of a given position[33]. Using this model, we compared the ratios of observed versus expected polymorphic sites in pG4s and their partitioned sections based on the genomic data of 503 individuals from the European subpopulation of 1000 genomes project (the EUR)[34].

The stability of a G4 structure depends on the lengths of loops and G-tracts, as well as nucleotide compositions on both sides of the G4[35]. We predict that the stability of promoter G4s may be under evolutionary selective pressure and contributes to promoter activity. Therefore, in the following analyses, we also considered the relative stability of pG4s as shown in Supplementary Fig. 1b.

Overall, the ratios showed a similar pattern between 1 kb and 0.2 kb promoters (Fig. 1h). First, the ratios in all- and stable-pG4s were higher than those in unstable-pG4 and the controls shown

as light blue horizontal shadows, which used the sequences of non-pG4 promoters with a 95% confidence interval (CI) (Fig. 1h), while stable-pG4s had higher ratios than all-pG4s. Second, for the G-tracts, the ratios in all and stable-pG4s were generally lower than those in unstable-pG4s and the controls, while stable-pG4s show lower ratios than all-pG4s. Third, for the loops, the ratios in all and stable-pG4s were higher than those in unstable-pG4s and the controls, while stable-pG4s showed higher ratios than all-pG4s. Fourth, for the regions upstream and downstream pG4s, and G-tracts in non-pG4 promoters, the ratios were similar to those of the controls, while their ratios among all-, stable- and unstable-pG4s were indistinguishable (Fig. 1h). Our data are consistent with a previous report that G4 motifs have higher mutation rates than intergenic regions in the human genome[19,36]. Importantly, we discovered that the high variation rates were attributed to the mutations in the loops, while the G-tracts are relatively conserved.

To assess the evolutionary pressure of pG4 sequences in promoters, we examined the evolutionary signature of promoter pG4s using the Hudson-Kreitman-Aquadé (HKA) test[37] that can be used in any genomic context to analyze evolutionary selection patterns of different genomic loci. This test typically compares the numbers of polymorphisms and fixed variants between two sets of regions, including a test group that is assumed to have evolved under selection and a control group with neutral evolution. In HKA analysis, we first used the abovementioned EUR DNA-seq data with >30× sequencing depth to get single nucleotide-polymorphic variants, and single nucleotide-fixed variants in human and orangutan genomes obtained by genome-wide alignment. Subsequently, evolutionary selectivity was assessed by calculating the odds ratios (ORs) of SNPs in pG4 sequences versus non-pG4 regions of pG4-containing promoters.

In 1 kb promoters, the ORs for all-, stable- and unstable-pG4s were greater than 1 (OR = 1.18 for all three, with $P < 1.2 \times 10^{-20}$, $1.58 \times 10^{-14}$, $3.30 \times 10^{-8}$, respectively) (Fig. 2a). In 0.2 kb promoters, the ORs of all and stable-pG4s were significantly greater than 1 (1.06, $P < 0.019$ and 1.06, $P < 0.044$, respectively), while unstable-pG4 was not (OR = 1.06, $p = 0.163$). For G-tracts in pG4s, the ORs are generally greater than 1 in both 1 kb and 0.2 kb promoters, while the ORs in stable- and unstable-pG4s were lower and higher, respectively, than all-pG4s. For loops, the results were mixed. In both 1 kb and 0.2 kb promoters, the ORs of loops in stable-pG4s were greater than 1 and also higher than all- and unstable-pG4s, while the ORs of loops in unstable-pG4 were less than 1 and lower than all- and stable-pG4s (Fig. 2a). These results indicate that the pG4s in both 1 kb and 0.2 kb promoters were under negative selection or balance selection, and the stability of pG4s in the 0.2 kb promoters positively impact the evolutionary pressure of G4 motifs.

The assessment of HKA depends on population size. Therefore, we further analyzed the SNPs from 2,504 individuals of the 1000 Genomes project with >30× sequencing depth to get single nucleotide-polymorphic variants. The distribution trend of ORs in pG4s and G-tracts of different pG4 groups versus non-pG4 regions in both 1 kb and 0.2 kb promoters (Fig. 2b) was consistent with that of the EUR in Fig. 2a. However, in both promoter groups, no significant difference was observed in the ORs of loops among different sets of pG4s, which could be attributed to the increased rare variants in a relatively large population. Noticeably, with increased population size, both all- and stable-pG4s showed reduced ORs, not significantly different from 1, while ORs in G-tracts remained the same pattern in 0.2 kb promoters. Next, we interrogated whether this observation could be extended to the alignments between human and other primates' genomes. In the UCSC database, the genomic data from 11 primate species are available and their hierarchical genetic relationship is displayed in

Fig. 2c. We individually analyzed the fixed single-nucleotide variants between the human genome and each of these 11 primate genomes by genome-wide alignments. The ORs for pG4s in 1 kb promoters were greater than 1 in most of the primate species, while, for pG4s in 0.2 kb promoters, most ORs were very close to 1 (Fig. 2d, e). Importantly, the ORs for pG4 G-tracts in 1 kb promoter were greater than 1 in all analyzed primate species, and pG4 G-tracts in 0.2 kb promoters also showed ORs >1 in 9 of these 11 species (Fig. 2f, g).

Overall, these results indicated that both pG4s and their G-tracts are under heightened selective pressure, and pG4 G-tracts are more powerful elements than entire G4 sequences in the HKA test to evaluate selection pressure on pG4s, which is due to the higher frequency of mutations in the loop regions of pG4s.

In principle, a significant HKA test result with an OR > 1 is consistent with either negative selection or balancing selection[38]. To determine which type of selection is applicable for pG4s, we used the Kolmogorov-Smirnov test to compare the site frequency spectra of pG4 sequences, G-tracts and loops versus non-pG4 sequences, in which we obtained significant HKA test results in 1 kb and 0.2 kb promoters. Importantly, we observed relatively high prevalence of polymorphic variants at low MAFs for pG4s, G-tracts and loops in both 1 kb and 0.2 kb promoters (Supplementary Fig. 6). These results suggested pG4s and G-tracts in the promoters evolved under negative selection.

We further verified our results using the Tajima's D test, a population genetic test statistic created by Fumio Tajima, which can be used to analyze whether a selected region is under selection pressure and distinguish directional from balancing selection[39]. A negative value of Tajima's D will suggest the presence of multiple low-frequency allele loci and occurrence of directional selection. Additionally, the Fu and Li's D and Fu and Li's F tests can also be used to evaluate whether a region is subject to either directional or balancing selection[40]. Therefore, we further used Tajima's D, Fu and Li's D and Fu and Li's F to assess the selective pressures on pG4s and G-tracts with different degrees of stability in 1 kb and 0.2 kb promoters. The Tajima's D values of the pG4s and G-tracts in 1 kb promoters and 0.2 kb promoters were less than -2, and the results of Fu and Li's D and F tests were also negative, indicating that both 1 kb and 0.2 kb promoters were subject to directional selection (Table 1).

To further analyze the selective pressure on pG4s in 1 kb and 0.2 kb promoters, we evaluated the selective pressure using a mutability-adjusted proportion of singletons (MAPS), which measures the relative enrichment of rare variants in specific types of variants that can explain difference in mutation rates based on local sequence contexts[41]. This method has previously been used to assess the degree of selective pressure on pG4s within UTRs in the individuals of the gnomAD database[17].

We only considered the SNPs in G-tracts consisting of three Gs and assessed the selection pressure of these Gs at different positions in a G-tract. Gs at central positions in G-tracts of 1 kb and 0.2 kb promoters were enriched with rare variants that are present in only one allele of the individuals in gnomAD (Fig. 3a, b). The enrichment of all variants in promoters showed MAPS scores similar to those of synonymous mutations in protein-coding regions of the genome, whereas central Gs in G-tracts of 1 kb and 0.2 kb promoters displayed a similar degree of selective pressure to that of missense mutations in protein-coding regions. Interestingly, the Gs at the center of G-tracts showed higher enrichment of single variants compared to the Gs at both ends of G-tracts in gnomAD. These results suggest that the first and third Gs of G-tracts in pG4s have lower selection pressure than that of the central Gs, probably due to the formation of noncanonical G4-like structures when either the first or third G is mutated[42].

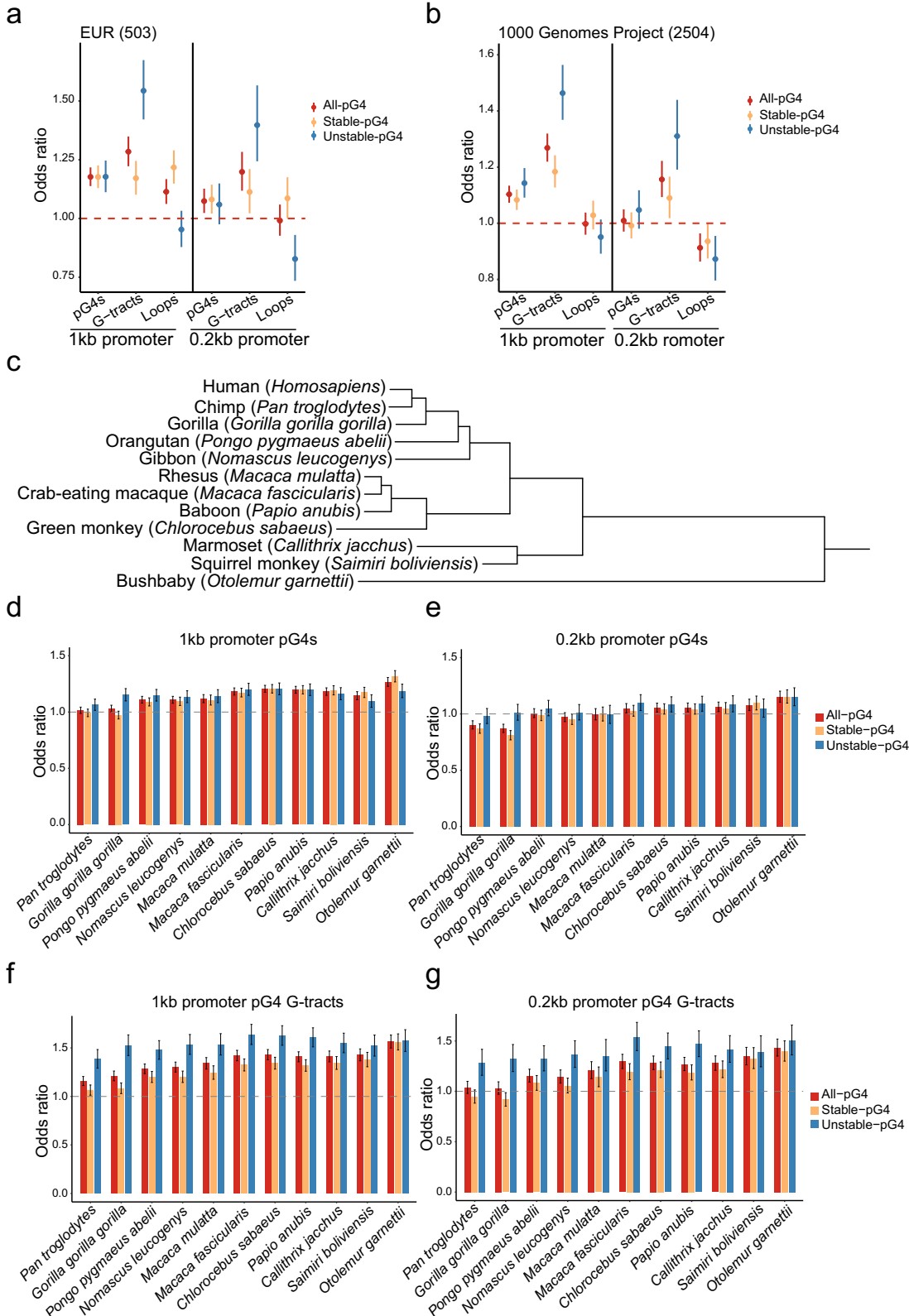

**Fig. 2 Odds ratios of Fisher's test to evaluate the significance of the Hudson-Kreitman-Aquade (HKA) in promoters.** The odds ratios (ORs) with 95% confidence intervals for 1 kb and 0.2 kb promoters in the EUR population containing 503 individuals (**a**) and 1000 Genomes project population containing 2504 individuals (**b**). "All-pG4s" suggests the sum of both stable-pG4s and unstable-pG4s. **c** Hierarchical clustering dendrogram to display genetic distance relationships among 12 selected primate species. **d**–**g** The ORs of pG4s in 1 kb (**d**) and 0.2 kb (**e**) promoters, and pG4 G-tracts in 1 kb (**d**) and 0.2 kb (**e**) promoters, in the 1000 Genomes project population. The error bar means 95% confidence intervals.

**Table 1 Tajima's D, Fu and Li's D and Fu and Li's F tests for 0.2 kb and 1 kb promoters.**

| Groups | Tajima's D | Fu and Li's D | Fu and Li's F |
|---|---|---|---|
| In 0.2 kb promoters | | | |
| G-tracts in stable-pG4s | −2.436 | −25.599 | −10.165 |
| G-tracts in all-pG4s | −2.428 | −26.137 | −10.289 |
| G-tracts in unstable-pG4s | −2.410 | −26.288 | −10.457 |
| Stable-pG4s | −2.346 | −25.070 | −9.876 |
| All-pG4s | −2.344 | −25.850 | −10.098 |
| Unstable-pG4s | −2.339 | −26.620 | −10.406 |
| In 1 kb promoters | | | |
| G-tracts in stable-pG4 s | −2.402 | −25.650 | −10.099 |
| G-tracts in all-pG4s | −2.391 | −26.040 | −10.193 |
| G-tracts in unstable-pG4 | −2.372 | −26.304 | −10.331 |
| Stable-pG4 | −2.301 | −24.846 | −9.738 |
| All-pG4 | −2.300 | −25.510 | −9.935 |
| Unstable-pG4 | −2.298 | −26.269 | −10.209 |

**Most pG4 sequences in promoters are isoform-restricted.** Over 10,000 protein-coding genes in human cells are regulated by two or more promoters. For each gene, the alternative use of different promoters can determine the position and length of its 5'-UTR, first exon, 3'-UTRs, and even CDS[43,44]. Unlike alternative splicing of genes, the selection and activity alteration of promoters are pre-transcriptionally regulated. Alternative promoters represent a major context- and tissue-specific mode of transcriptional regulation and contribute to transcript diversity. In many cases, the overall expression of a cancer-relevant gene remains unchanged in cancer cells versus their normal counterparts, but its different promoters may show distinct activities, leading to increased expression of cancer-promoting transcripts. Therefore, compared to differential gene expression, promoter activity can be more accurate in predicting the clinical outcomes of cancer patients[44]. We hypothesized that pG4 sequences in promoters may affect the selection and activity of promoters, based on their selective constraints. Although G4 motifs in the promoters of many oncogenes, such

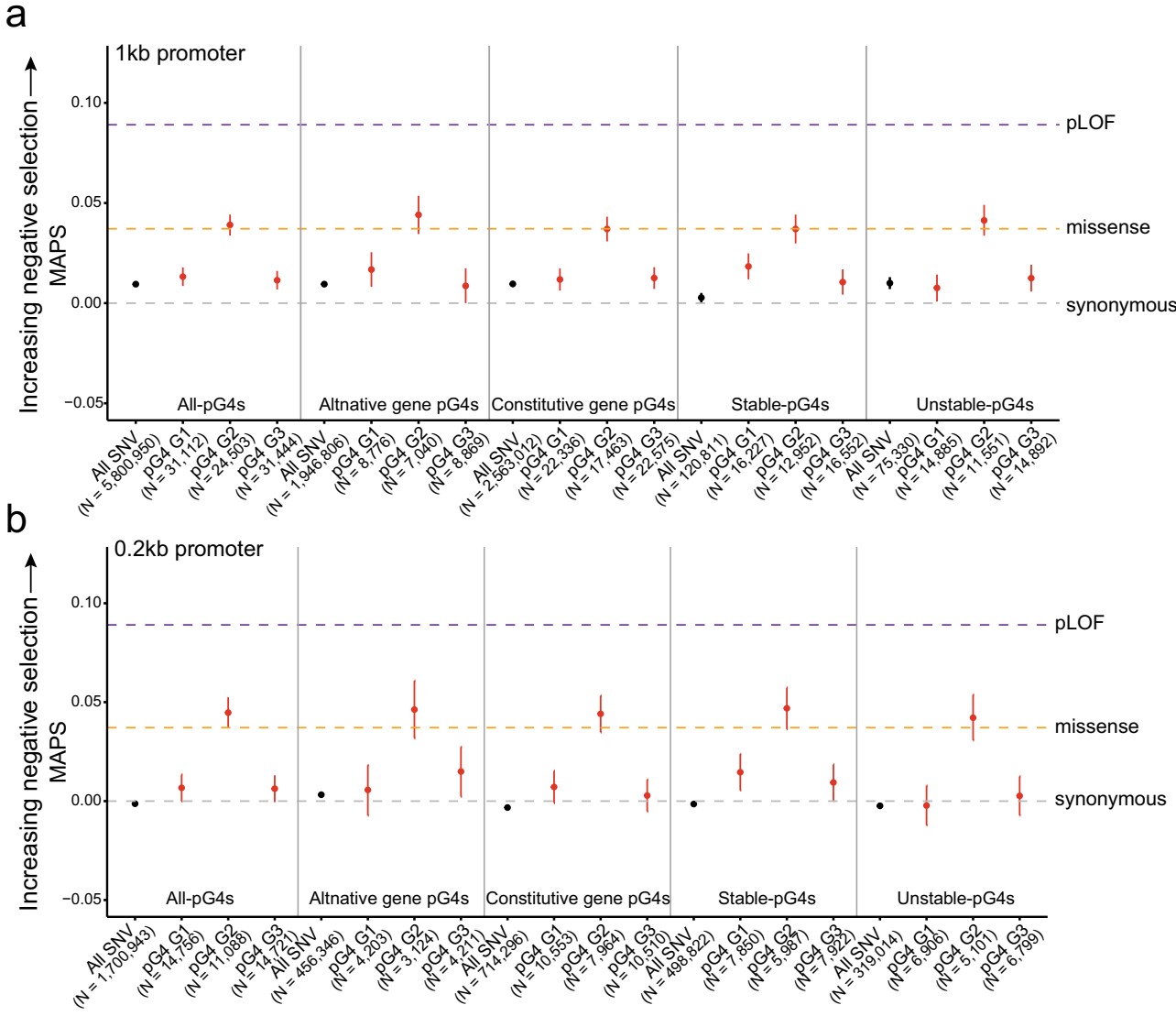

**Fig. 3 MAPS scores for pG4s and their different sections or G positions in 1 kb and 0.2 kb promoters. a, b** Mutability-adjusted proportion of singletons (MAPS) for each set of variants affecting the three guanines (Gs) within pG4 sequences in 1 kb promoters (**a**) and 0.2 kb promoters (**b**). Error bars represent the 5% and 95% bootstrap permutations for each variant class. The three dash lines in purple, orange and gray depict the MAPS scores for Ensembl predicted high-impact coding (i.e., predicted loss-of-function), missense and synonymous mutations, respectively.

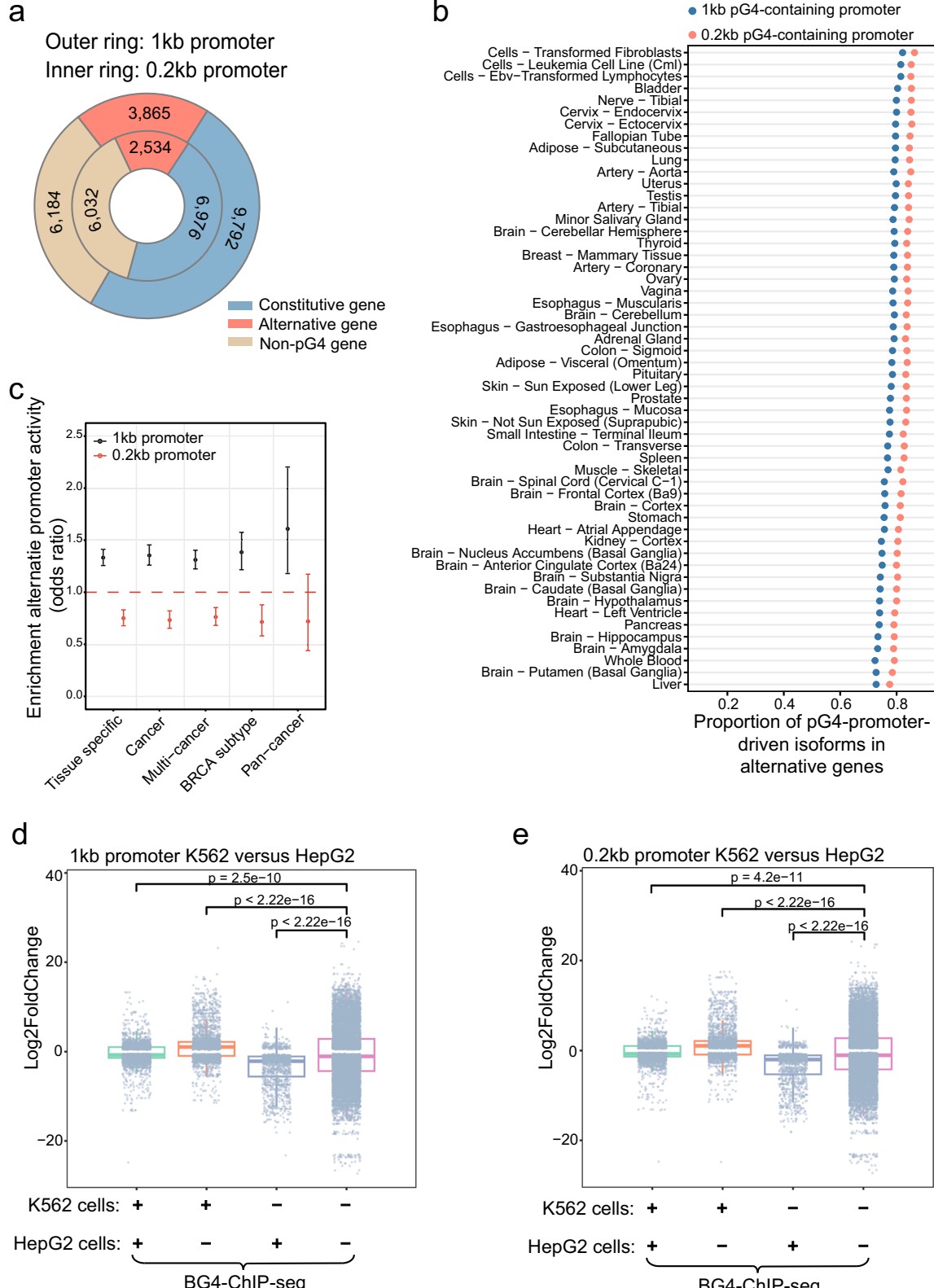

as *MYC*, *KIT* and *KRAS*, could reportedly inhibit their gene expression, it remained unclear whether G4s in all or most promoters had similar regulatory activity.

We define the genes that have at least one pG4 in each of its annotated promoters as constitutive genes, and the genes that have both pG4-containing and non-pG4 promoters as alternative genes. For the majority of constitutive genes, different promoters

of each gene shared the same pG4 sequence(s) (8206 of 9792 genes for 1 kb promoters, and 6163 of 6976 genes for 0.2 kb promoters); meanwhile, 1586 and 813 constitutive genes for 1 kb and 0.2 kb promoters, respectively, harbored different pG4s (Fig. 4a). Alternative genes account for about 28.3% (3865 out of 13,657) and 26.6% (2534 out of 9510) of the genes with pG4s in 1 kb and 0.2 kb promoters, respectively.

**Fig. 4 Distribution and usage of transcript isoforms with pG4s in 1 kb and 0.2 kb promoters. a** Distributions of constitutive, alternative and non-pG4 genes defined by 1 kb promoter (outer ring) and 0.2 kb promoters (inner ring). Constitutive genes are those with each promoter containing pG4 sequences, alternative genes are those with at least one promoter containing a pG4 sequence(s), and non-pG4 genes are those without any pG4-containing promoter. **b** The proportions of transcript isoforms driven by pG4-containing 1 kb and 0.2 kb promoters in alternative genes. To calculate the proportion of the transcripts driven by pG4-containing promoters for an alternative gene, the numbers of the isoforms produced by pG4-containing promoters were divided by the numbers of total transcripts driven by both pG4-containing promoters and non-pG4 promoters. The median expression (TPM) of each transcript with or without the pG4 promoter was assessed in different tissues, and a transcript with its TPM in a tissue greater than 1 was considered as being expressed. **c** The odds ratios of the pG4-containing promoters from five datasets including tissue specific cancers, all cancers, multi-cancers, BRCA subtype specific tumors and pan-cancers obtained from a published work[44]. Those pG4-containing promoters belong to the genes that showed no difference of overall gene expression among different tumor tissues (for "tissue specific" dataset) or between tumor and normal tissues (for other four datasets), but had differential activity among different promoters of each gene. Comparison of individual 1 kb (**d**) and 0.2 kb (**e**) promoter activities between K562 and HepG2 cell lines based on RNA-seq data. The promoters are put into 4 groups according to their G4 structure statuses determined by BG4-ChIP-seq analyses. The error bar means 95% confidence intervals. Each box plot (central line, median, box limits, upper and lower quartiles) represents the log2-fold-change on each group.

MAPS scores for both alternative and constitutive pG4 promoters showed that the central Gs were enriched with more rare variants ($P < 0.05$) than the first and third Gs, and showed similar scores to those of missense variants (Fig. 3a, b). This suggests that the central Gs of the G-tracts in pG4s of constitutive and alternative promoters exhibit a similar pattern of selective confinement.

We next evaluated whether expression of the transcripts from alternative genes is restricted in different tissue environments. Using transcript expression data from 54 different tissues collected by the GTEx database[45], we observed that, in alternative genes, the isoform transcripts driven by pG4-containing 1 kb and 0.2 kb promoters occupied 72.4–82.1% and 77.6–86.4% of all transcripts, respectively (Fig. 4b). Additionally, we found that about half of the promoters in alternative genes contained pG4s (Supplementary Fig. 7). Overall, our data suggest that G4 motifs in the promoters of alternative genes contribute to the selective expression among different transcripts and are positively associated with enhanced promoter activity versus non-pG4 promoters.

Demircioglu, et al. reported that the majority of genes in tumor samples from different tissue types exhibited no significant difference in their overall expression versus the controls of matched normal tissues, but showed different levels of individual transcripts, suggesting distinct activity of individual promoters[44]. We conducted the following analyses to evaluate whether pG4s regulate the transcriptional activity of promoters. From a published work[44], we obtained five available datasets for tissue specific cancers, all cancers, multi-cancers, BRCA subtype specific tumors and pan-cancers, and extracted the data of the alternative genes with no change in overall expression, but significantly different activity among individual promoters. After comparing the proportions of transcripts driven by pG4-containing alternative promoters versus those transcribed by all other promoters, we found that the ORs of the 1 kb alternative promoters from all five datasets were greater than 1, while the ORs of the 0.2 kb alternative promoters from the first four datasets were less than 1, and OR of pan-cancer associated 0.2 kb alternative promoters was not significantly different from 1 (Fig. 4c). Overall, these results indicated that pG4-containing 1 kb promoters are major contributors to differential promoter activities of alternative genes among different tumor tissues, or between tumor and normal tissues, due to their OR > 1 (Fig. 4c). In contrast, pG4-containing 0.2 kb promoters contribute to maintaining relatively comparable or stable promoter activities (Fig. 4c).

We further analyzed the effects of G4s on promoter activity by comparing the BG4 ChIP-seq data and RNA-seq data in K562 and HepG2 cells[28]. We calculated the ratios of promoter activity in K562 versus those in HepG2 cells after dividing them into four groups according to the presence or absence of G4s in their promoters identified by BG4 ChIP-seq analyses. These four groups included G4 positive in both cell types, only in HepG2 cells, only in K562 cells and in neither cell type, and the ratios were determined as 0.98, 0.49, 1.26 and 1.02, respectively, for 1 kb promoters (Fig. 4d), and 0.98, 0.48, 1.27 and 1.01, respectively, for 0.2 kb promoters (Fig. 4e). We further analyzed the RNA-seq and ChIP-seq data of additional two pairs of cell lines, U2OS versus K562 and A549 versus H1975, and the four promoter groups in both pairs showed the same promoter activity pattern to that between K562 and HepG2 cell lines (Supplementary Fig. 8). Based on the ratios either significantly larger or smaller than 1, we concluded that the absence of G4s could reduce promoter activity, while the presence of G4s could increase promoter activity. Thus, our results support the hypothesis that G4 structures in both 1 kb and 0.2 kb promoters enhance promoter activity.

G4 ligands can stabilize G4 structures, which mostly leads to gene repression[3]. As a G4 stabilizer, TMPyP4 effectively inhibits cancer cell proliferation and is thus a potential anti-cancer agent[46]. Li, et al. combined the BG4 antibody and CUT&Tag approach to create the G4-CUT&Tag method, and used it to identify native G4s in the genome of cells treated by TMPyP4 versus those in solvent treated cells[30]. In this report, the authors also used transient transcriptome sequencing (TT-seq), an RNA sequencing approach that can quantify nascently transcribed RNA, to determine TMPyP4-caused changes in gene expression. We used the G4-CUT&Tag and TT-seq datasets from this study (GSE178668) to interrogate the relationship between G4 formation and gene expression. Among 23,674 tested genes, 2135 were downregulated in response to TMPyP4 treatment versus the control, but no upregulated gene was detected (FDR < 0.05) (Fig. 5a). However, further analyses indicated that TMPyP4 treatment could only change the activity of 9 promoters. TMPyP4 downregulate many genes, but only a minor difference in promoter activity was observed, which may be due to the fact that TT-seq could only detect nascent RNA fragments within 5 min.

Interestingly, when analyzing G4 occupancy, we observed an increase of 1297 G4 signals after TMPyP4 treatment versus the control, without any detectable G4 signal loss at other loci (Fig. 5b). Importantly, 92.6% of these increased G4 signals were distributed in promoter regions, which likely contributed to TMPyP4-mediated gene downregulation. Our further analyses revealed that 252 genes showed both G4 signal increase and reduced expression in response to TMPyP4 treatment (Fig. 5c). After functional enrichment analysis using the Gene Ontology (GO) annotations for the category of biological processes, we discovered that these 252 genes were mainly involved in histone

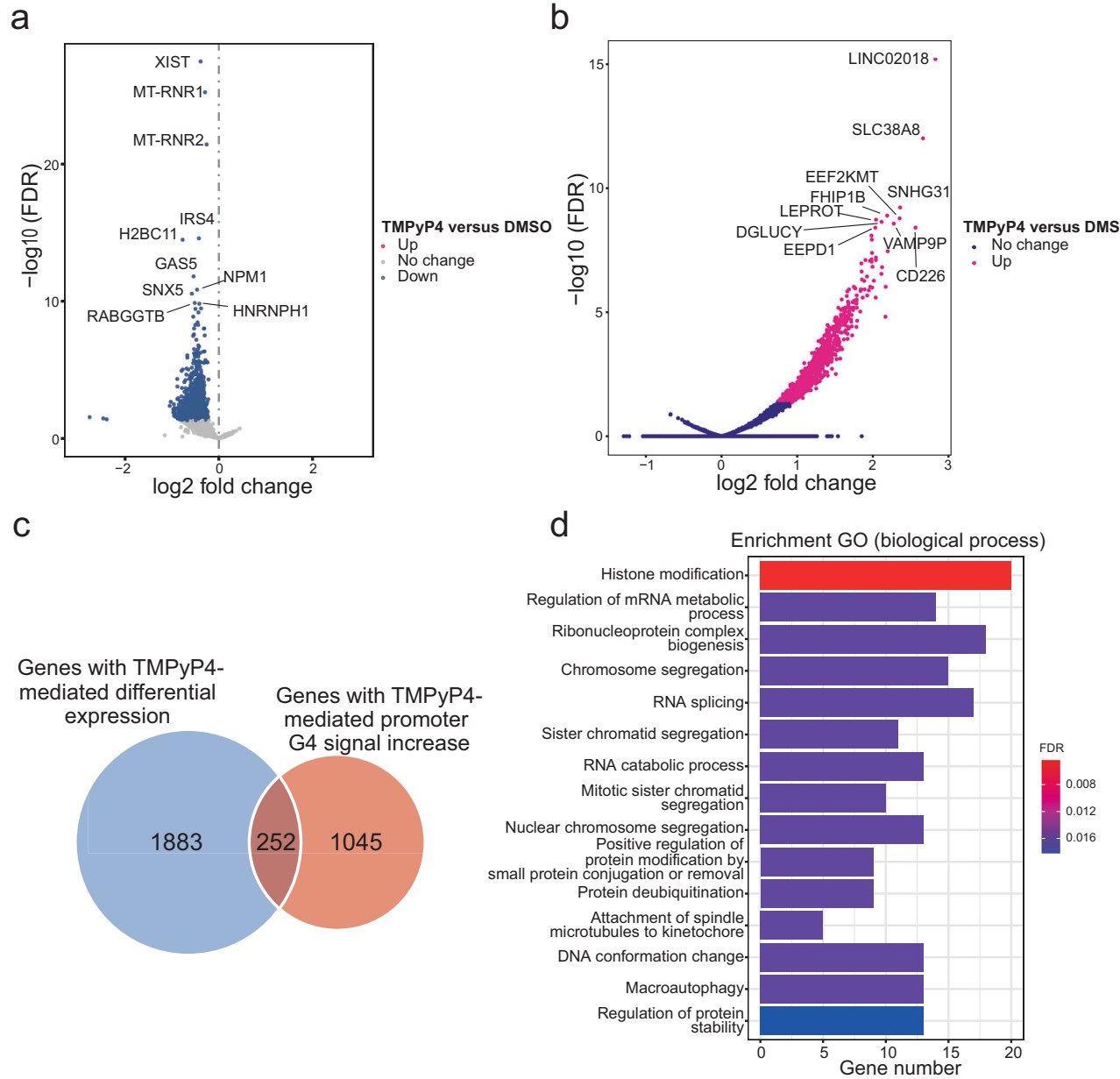

**Fig. 5 Alterations of G4 signals and gene expression in TMPyP4-treated HEK293T cells. a** Volcano map of differentially expressed genes in TMPyP4-treated HEK293T cells. The gene expression data was obtained from a previous study[30] that used the TT-seq approach to analyze the transcripts of HEK293T cells treated by TMPyP4 and DMSO. **b** Volcano map of differential enrichment of G4 signals obtained by G4-CUT&Tag approach in TMPyP4-treated HEK293T cells. **c** Venn diagram of the overlapped genes between annotated genes enriched by G4-CUT&Tag identified G4 signals and differentially expressed genes determined by TT-seq in TMPyP4-treated HEK293T cells. **d** GO functional enrichment analysis for the category of biological processes of the overlapped genes identified in **c**.

modifications, mRNA metabolism, chromatin segregation, and RNA splicing, among others (Fig. 5d). The results imply that many genes participating in various epigenetic processes are regulated by G4 motifs.

**pG4s in promoters are enriched for *cis*-eQTLs.** Expression quantitative trait loci (eQTLs) represent the SNPs in genome associated with gene expression levels, while eQTLs located near the gene-of-origin are defined as *cis*-eQTLs[47]. To evaluate the contributions of G4 motifs in promoters to gene expression, we analyzed the enrichment of *cis*-eQTL-associated SNPs in pG4s versus non-pG4 sequences in both 1 kb and 0.2 kb promoters. We

categorized the promoter *cis*-eQTLs into four groups, including Lead (SNPs with the most statistically significant changes in each tissue), Causal (SNPs showing causal relationship with gene expression when screened by CaVEMaN, a computational model)[48], TF (SNPs residing in any binding site of 655 transcription factors), and Nominal (all SNPs with statistically significant changes). With the *cis*-eQTLs in non-pG4 sequences of 1 kb promoters as controls, we observed that all four groups of *cis*-eQTLs were highly enriched in the pG4s of 0.2 kb promoters, while the first three *cis*-eQTLs, but not the Nominal, were also in enriched in the pG4s of 1 kb promoters, although their corresponding enrichments were markedly lower than those in 0.2 kb promoters (Fig. 6a). Furthermore, we observed similar enrichment pattern of

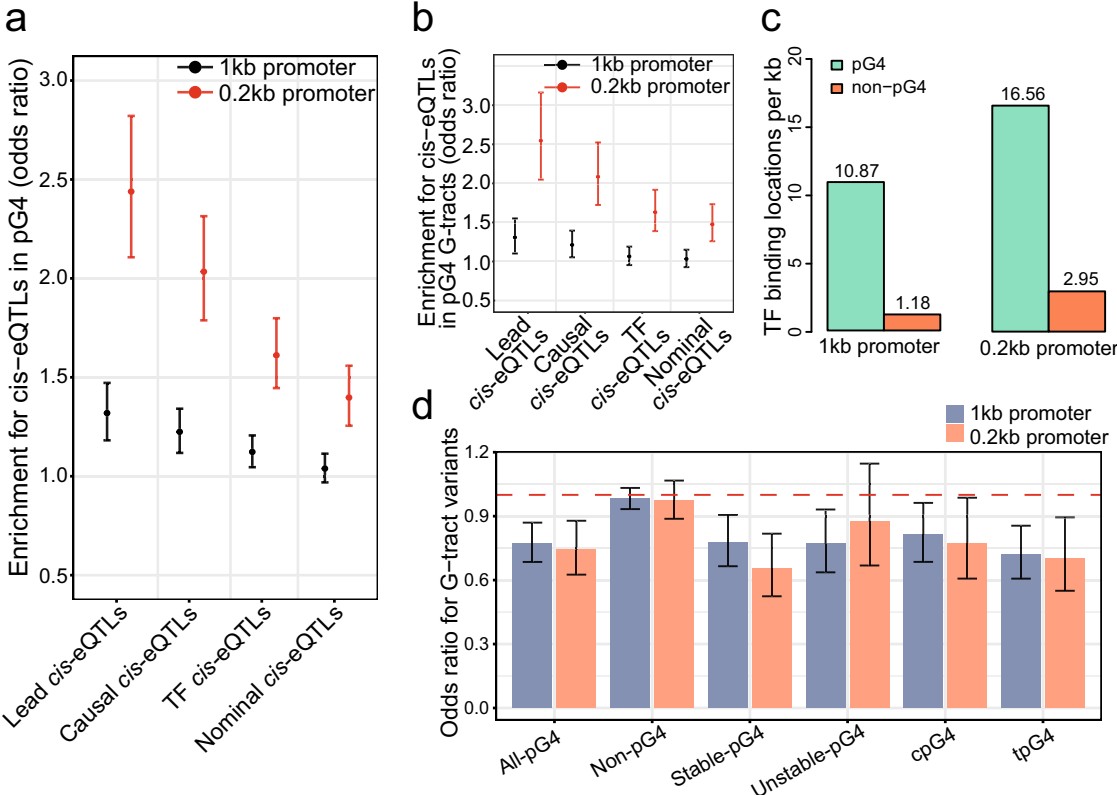

**Fig. 6 Enrichments of *cis*-eQTLs in pG4s and G-tracts in the pG4-containing promoters.** Enrichments of differentially defined *cis*-eQTLs in pG4s (**a**) and G-tracts (**b**) in pG4-containing promoters. The *cis*-eQTLs in pG4-containing 1 kb and 0.2 kb promoters from the GTEx database were categorized into four groups, including Lead (SNPs with the most statistically significant changes in each tissue), Causal (SNPs showing causal relationship with gene expression when screened by CaVEMaN, a computational model)[48], TF (SNPs residing in any binding site of 655 transcription factors), and Nominal (all SNPs with statistically significant changes). The error bars represent the 95% confidence interval for odds ratio. Lead eQTLs represents the *cis*-eQTLs with the most significant differences in each gene in each tissue. Causal eQTLs indicates *cis*-eQTLs with causal relationship using CaVEMaN methods. Nominal eQTLs represents *cis*-eQTLs affecting gene expression. TF eQTLs represents *cis*-eQTLs in the transcription factor recognition region. **c** Comparison of the overlap between pG4s and TF binding sites in pG4-containing and non-pG4 promoters. Alignment between pG4s, or random regions in non-pG4 in promoters, and the interaction sites of 655 TFs based on their ChIP-seq data in HepG2 and K562 cells was carried out. The results are presented as the numbers of TF binding sites in 1 kb of analyzed sequences. **d** The odds ratios of gene expression difference caused by G-tract variations in different types of pG4s in promoters. The gene expression caused by variations in G-tracts versus loops in different pG4s was evaluated, using the data of G-tracts in non-pG4 versus non-G-tract sequences is used for comparison. The error bar means 95% confidence intervals.

different *cis*-eQTL groups in G-tracts of pG4s versus those of non-pG4s, except that OR values of TF and Nominal *cis*-eQTLs in 1 kb promoters were indistinguishable from 1 (Fig. 6b). These results indicated that pG4 and G-tract sequences are important *cis*-regulatory elements of gene expression.

Next, to evaluate the potential regulatory roles of G4 motifs in gene expression, we analyzed the overlapping of pG4s with the interaction sites of 655 TFs based on their ChIP-seq data in HepG2 and K562 cells obtained from ENCODE database. We discovered 10.87 and 1.18 TF binding sites in each kb of sequences for both 1 kb pG4 and non-pG4 promoters, respectively, while for 0.2 kb pG4 and non-pG4 promoters, 16.56 and 2.95 TF binding sites per kb of sequences were identified, respectively (Fig. 6c). suggesting that G4 structures are intensively involved in TF-mediated gene expression, especially in 0.2 kb promoters. Additionally, we analyzed gene expression difference caused by variations in G-tracts versus loops in different pG4s, using G-tracts in non-pG4 versus non-G-tract sequences for comparison. Based on the OR values, changes in all-pG4s, stable-pG4s, cpG4s and tpG4s could significantly decrease the activity of both 1 kb and 0.2 kb promoters, while variations in unstable-pG4s and non-pG4s showed insignificant effect (Fig. 6d and Supplementary Data 1), reinforcing the notion that G4s play a positive role in gene expression.

**Promoter G4s overlap with various gene expression regulatory elements.** The binding motifs of many TFs, such as SP1, SP2, MAZ and E2F4, contain G- or C-tracts, suggesting that their binding sites may overlap with pG4s, and their DNA binding affinity can also be regulated by G4 formation[28]. Zyner, et al. reported that G4-containing promoters were enriched with H3K4me3 at different differentiation stages of embryonic stem cells[49]. Interestingly, based on our analysis, TMPyP4-repressed genes were also enriched in those that regulate histone modifications (Fig. 5d), suggesting that the presence of G4s in promoters plays a causal role in histone modifications. To quantitatively evaluate the association of G4s with histone statuses, we analyzed the enrichment of 11 histone modifications and the histone variant H2AFZ, due to the availability of their ChIP-seq data in ENCODE, in promoter pG4s and promoter G4 structures identified by BG4 ChIP-seq in K562 and HepG2 cells. Multiple histone marks of gene activation, including H3K4me2/3, H3K27ac and H3K9ac, were enriched in the G4s of both 1 kb and 0.2 kb promoters based on both predicted pG4s and BG4-recognized G4s in the two cell lines (Fig. 7a–d), suggesting that G4 motifs are present in promoter regions associated with active gene expression. Interestingly, H2AFZ, a histone H2A variant produced by one of the two *H2A.Z* genes, was also

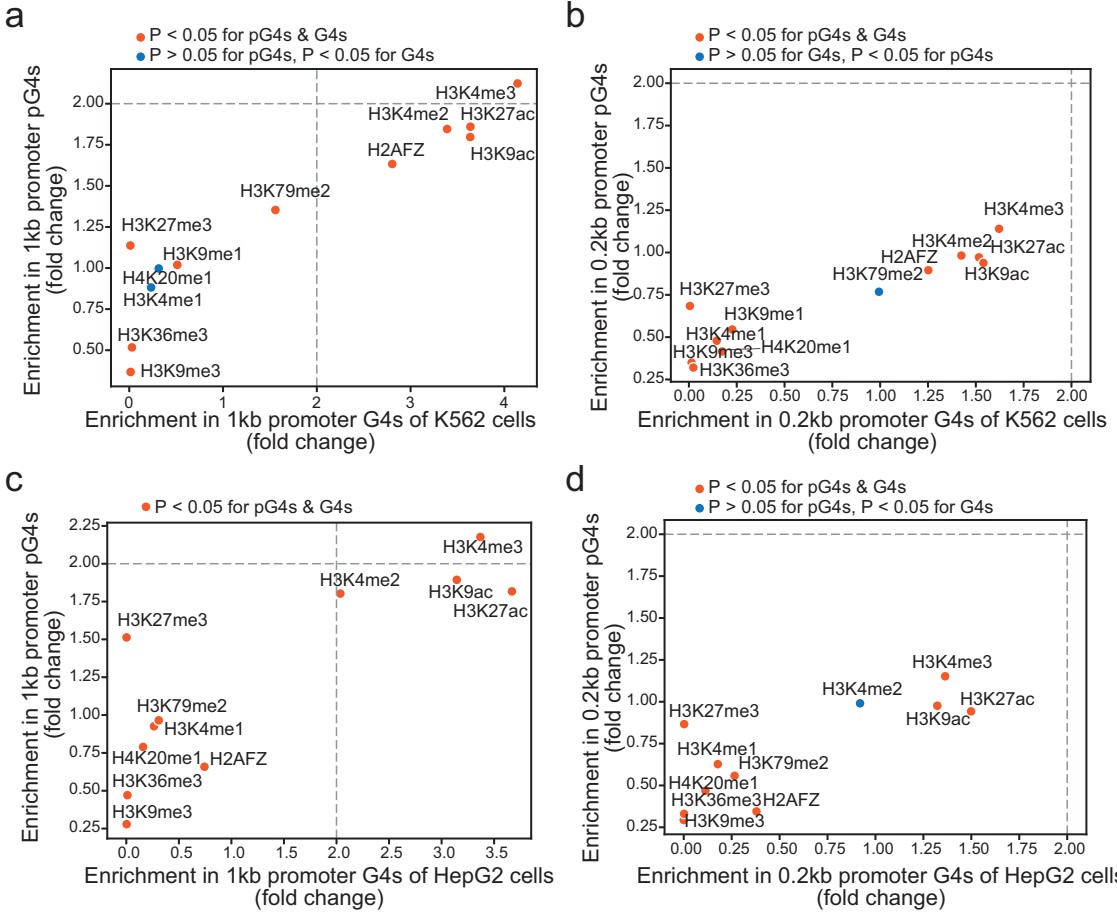

**Fig. 7 Correlation of pG4s and experimentally identified G4s with histone modifications and a histone variant in promoters. a–d** Scatter plots of pG4s and BG4-ChIP-seq identified G4s versus 11 histone modification marks and a histone H2A variant H2AFZ in K562 and HepG2 cells based on the ChIP-seq data obtained from the ENCODE database. **a** and **b** are the graphs of 1 kb and 0.2 kb promoters in K562 cells, and **c** and **d** are the graphs of 1 kb and 0.2 kb promoters in HepG2 cells. Correlations with *P* < 0.05 and *P* > 0.05 are presented in orange-red and blue color dots, respectively.

enriched in G4-containing promoters, especially in K562 cells. H2AFZ has been reported to be overexpressed in multiple cancers, and positively regulate cancer cell proliferation and metastasis[50]. Noteworthily, H3K9me3, a reputed gene repression mark exhibited less than 0.5 fold change in 1 kb and 0.2 kb promoters in K562 and HepG2 cells (Fig. 7a–d).

Next, we carried out the same analyses to evaluate the enrichments of the interaction sites of 85 chromatin remodeling proteins and 655 TFs in promoter pG4s, with non-pG4 sequences in corresponding regions as controls. In K562 cells, the interaction sites of many chromatin remodelers and TFs were significantly enriched in the sites of both pG4 sequences and BG4-recognized sequences in 1 kb and 0.2 kb promoters (Fig. 8a–d). Meanwhile, their enrichments in the sites defined by both pG4 prediction and BG4 ChIP-seq in the promoters showed remarkably positive correlations. However, based on the analyses of the data derived from HepG2 cells, these positive correlations were markedly reduced, even showing negative correlation coefficients in some cases, although the enrichments of most of these chromatin remodeler and TF binding sites in both pG4s and BG4-recognized G4s were still statistically significant (Supplementary Fig. 9).

Noteworthily, the fold changes (FCs) of the BG4-recognized promoter regions enriched by the chromatin remodelers and TFs were remarkably higher than those of the enriched sites in the promoter pG4s (Fig. 8 and Supplementary Fig. 9). The results strongly suggest that BG4 recognized motifs, or G4 structures,

likely coexist with chromatin remodelers and TFs in promoters and play critical roles in gene expression.

## Discussion

In the current study, we applied a comprehensive catalog of human genetic variations to assess evolution pressure on G-quadruplex formation sequences in promoters. We hypothesized that pG4 sequences would exhibit relatively low variations if they were biologically functional. Supporting this hypothesis, we show that variations in 1 kb and 0.2 kb promoters are reduced compared to non-pG4 promoters with the same lengths using a local sequence context based on substitution models. We used Tajima's D, Fu and Li's D and F, and HKA tests to assess the evolution pressure on G4 motifs and observed that pG4 in 1 kb and 0.2 kb promoters were under the negative selection. Interestingly, when we used HKA test to evaluate the evolutionary pressure of pG4s in promoters, the ORs obtained from the EUR population with 503 individuals were consistent with the results reported previously[21]; however, based on the analysis of the variations of all populations in the 1000 genomes comprised of 2504 individuals, we did not find significant difference between non-G4 regions and pG4 sequences in 0.2 kb promoters (Fig. 2b). Importantly, the same analyses revealed that the evolutionary pressure in pG4 G-tracts of both 1 kb and 0.2 kb promoters were highly significant in both populations. When extending our analyses to the genome-wide alignments between human genome

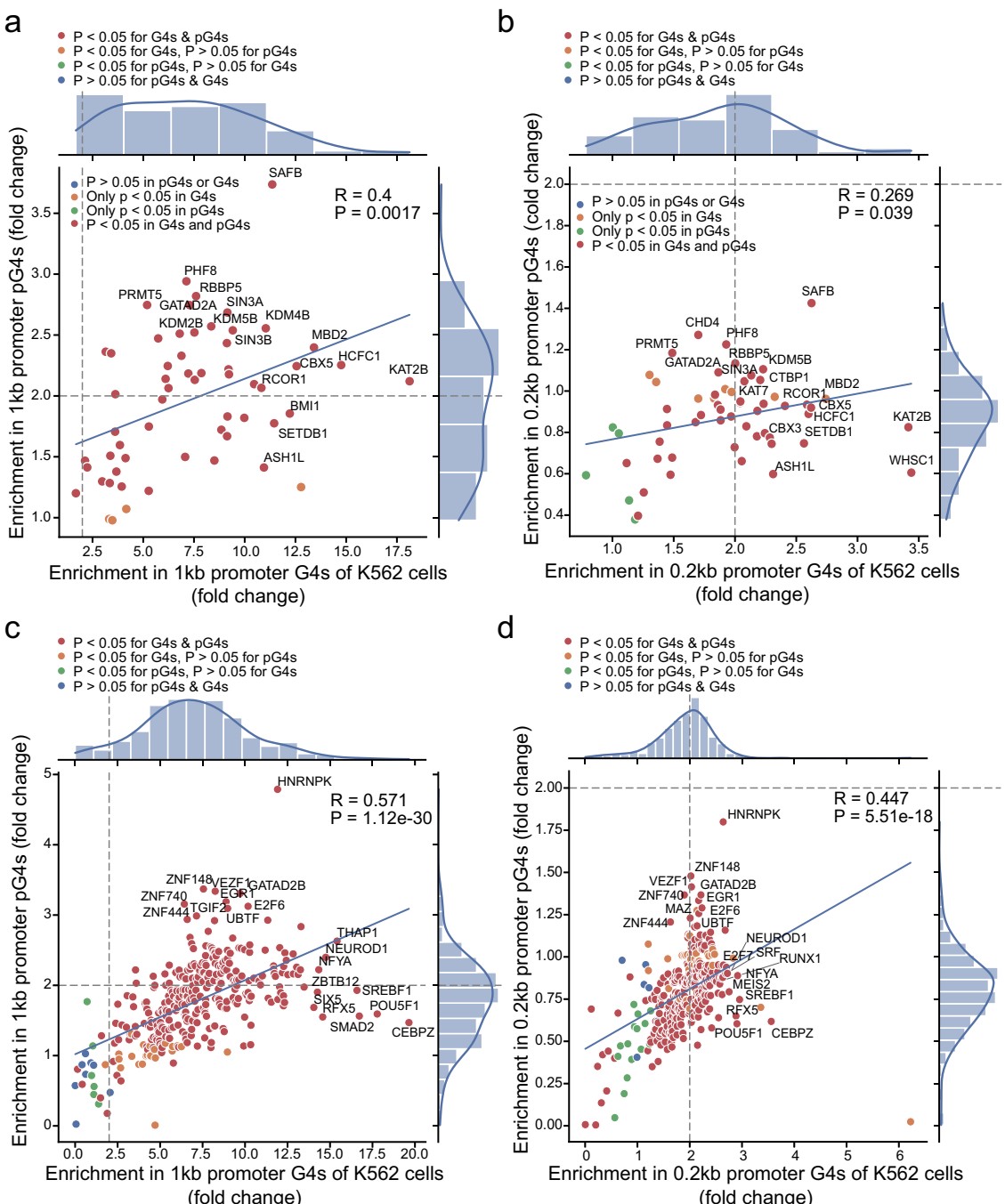

**Fig. 8 Correlation of pG4s and experimentally identified G4s with chromatin remodeling proteins and TFs in promoters in K562 cells. a–d** Scatter plots of pG4s and BG4-ChIP-seq-identified G4s versus 59 chromatin remodeling proteins (**a** and **b**) and 338 TFs (**c** and **d**) in 1 kb promoters (**a** and **c**) and 0.2 kb promoters (**b** and **d**) in K562 cells based on the ChIP-seq data obtained from the ENCODE database. Correlations with $P < 0.05$ and $P > 0.05$ are presented by different color dots as indicated. The overall correlation coefficient and P value of each graph are provided.

and each of 10 additional primate genomes, we also observed that pG4 G-tracts are under relatively high selective pressure compared to pG4s (Fig. 2d–g). Therefore, our data strongly suggest that G-tracts are more powerful than pG4s to evaluate the revolutionary pressure of G4 structures. The underlying reason is likely that G-tracts are the most essential elements in G4 structures. A previous study indicated higher mutation ratios in pG4s than those in non-pG4 regions of promoters[36]. Therefore, we further analyzed the tendency of variations in different partitioned sections of pG4s and discovered that the substitution ratios of G-tracts were remarkably lower than those of connecting

loops among G-tracts (Fig. 1g, h). Therefore, our data support the notion that the high mutation propensity of loop regions contribute to the high substitution ratios of pG4s, which causes genomic instability.

Our analysis of positional constraint within the pG4 sequences revealed that the selective pressure exerted on the central Gs of G-tracts containing only 3 Gs was comparable to that of missense mutations in protein-coding regions of the genome. Consistently, central G alterations of pG4s exhibited the most deleterious effects on biological functions of telomeres and UTRs[17,18]. Furthermore, the central G was more constrained than the other two

Gs in a G-tract, possibly due to the formation of noncanonical G4 structures when the first or third G in a G-tract was mutated. In line with this assumption, in the *PDGFRB* promoter, guanine-metabolized dGMP could be used in the first G position of a G-tract to form G4 structure[51].

The enrichment of eQTLs in a region implies its high potential as a crucial regulatory element(s) in gene expression. We have mapped a significantly greater amount of *cis*-eQTLs in pG4s than that in non-pG4 sequences of 1 kb and 0.2 kb promoters (Fig. 6A), suggesting that promoter G4s have high potency in regulating target gene expression. Importantly, comparison between three pairs of cancer cell lines revealed that G4 structure-containing promoters in one cell line showed significantly higher activities than corresponding promoters without G4 formation in another cell line, and vice versa (Fig. 4d, e and Supplementary Fig. 8). Consistently, changes in G-tracts of promoter pG4s, especially those in stable-pG4s and tpG4s, significantly reduce the transcript levels of target genes (Fig. 6d and Supplementary Data 1), supporting the stimulative role of promoter G4s in gene expression.

Despite the association of G4s with active gene expression observed in our analyses and in a number of other omics-based studies, both positive and negative regulatory effects of promoter G4s on a variety of cancer-related genes have been reported. As examples, the expression of *BCL2* and *KRAS* was activated by the G4s in their promoters[52,53], while *OCT4* and *ARID1A* were repressed[54,55]. The discrepancy among these reports could be due to the following reasons. First, many early studies employed plasmid-based reporter assays to evaluate the effects of G4s on gene expression, which obviously could not reflect the behavior of G4s in a genomic context. Second, in these studies, the judgment of G4-mediated regulation was based on the effects of G4 stabilizers on gene expression, which could easily lead to the conclusion that G4 formation would cause gene repression. In our analyses, TMPyP4 could repress a large number of genes without upregulating any gene (Fig. 5a). Third, promoter G4s may have differential regulatory activities and display distinct effects on the expression of different genes. Their regulation can also be dynamic and variable depending on physiological and pathological conditions.

TMPyP4's exclusively inhibitory effects on gene expression suggest that ligand binding may hamper the association or assembly of transcription machinery on target gene promoters. Further analysis revealed that 252 DEGs repressed by TMPyP4 were mainly the genes regulating various epigenetic events, especially histone modifications (Fig. 5c, d), which was consistent with the observation of several previous studies[30,56,57]. Concurrently, our analyses also demonstrated the enrichment of histone modification marks, and the interaction sites of chromatin remodeling factors and TFs in both pG4s and ChIP-seq-identified G4s within 1 kb and 0.2 kb promoters. These results implicate that G4s may serve as regulatory hubs for differential and dynamic chromatin activities. As the most enriched histone modifications are the well-characterized gene activation marks, such as H3K4me2/3, H3K27ac and H3K9ac (Fig. 7), our data strongly support the notion that promoter G4s are formed in nucleosome-depleted and transcriptionally active genomic regions, and play a positive role in gene expression, consistent with the results of previous reports[27,56,58].

The current study has the following limitations. First, we applied a text-based approach toward identifying regions of potential G4 structures in promoters. Although this approach has been frequently employed in previous work[12,59], certain sequences that are inconsistent with the canonical G4 motifs may also form G4-like structures[24,60]. Given very limited evidence that many of these noncanonical G4s are readily formed *in cellulo*, we

used a relatively stringent definition of G4 motifs, but alternative G4 sequences or structures could have been missed in our analyses. Thus, our assessment of sequence constraints and functional enrichment within the promoter G4-forming region may be incomplete. Second, although we have found evidence that G4 structure formation is restricted, whether these pG4s form secondary structures *in cellulo* is unclear. Third, our assessment of selective pressure exerted on pG4s in promoters using MAPS measures is limited by a relatively low number of variants. Fourth, although our data strongly support that G4s positively regulate promoter activity, experimental validation for these findings is needed in future investigations.

Overall, our study demonstrates that promoter pG4 sequences are under negative selection and important *cis*-regulatory elements in promoters to generally activate gene expression.

## Methods

**Identification of G-quadruplex sequences in 1 kb and 0.2 kb promoters**. TSS is defined by the transcript start site of the basic isoform of each gene in Genecode (v38) that uses the human genome hg38. The 1 kb promoter is defined by the 1 kb sequence upstream of the TSS of each transcript. The 0.2 kb promoter or active promoter is defined by ENCODE[25] and overlaps with the 200 bp upstream of the TSS. Putative G-quadruplex forming sequences (pG4) and genomic coordinates were identified by Quadron software[7] using its default arguments. Over 75% overlap of pG4 and promoter is defined as promoter pG4. 1 kb promoters, 0.2 kb promoters, pG4s and genomic coordinates were obtained using a custom python script. This approach yielded a total of 26,247 genes with 64,311 pG4s in 1 kb promoters and 12,328 genes with 20,764 pG4s in 0.2 kb promoters.

We defined the pG4s with predicted stability scores of >19 as stable-pG4, and those with scores ≤19 as unstable-pG4. All these pG4s were designated as all-pG4s.

**Variant frequency analyses**. Variants from the gnomAD release 3.1 were obtained (URL: https://gnomad.broadinstitute.org/downloads), and filtered to exclude those marked with low complexity regions and segmental duplication regions. Thus, only those variants that were identified by the FILTER as PASS with a mutation type of SNP were reserved. Those filters were generated by bcftools (version, 1.9). Additionally, only the variants from the alleles with their sequences clearly identified in over 80% of the whole cohort (76,156 individuals) were selected for analyses, which could reduce the variations caused by sequencing depth in the gnomAD v3.1 dataset[32]. The obtained dataset of variants with relatively high confidence was further analyzed by bedtools (version 2.30.0) intersected with the -u and -b flags through overlapping the dataset with the genomic coordinates for 1 kb promoters, 0.2 kb promoters, 1 kb promoter pG4s, 0.2 kb promoter pG4s, CDSs and non-pG4 regions.

A match set of sequences for non-pG4 promoters with constrained transcript-levels according to gnomAD observed versus expected features of the 1 kb promoters and 0.2 kb promoters separately was randomly selected from a protein-coding transcript constraint set, which was generated from the variations in 141,456 humans[32] (https://gnomad.broadinstitute.org/downloads#v2-constraint). This process consisted of transcript constraint matching between pG4 and non-pG4 sequences according to the information of loss of function variants metric (LOEUF) for each transcript (Supplementary Fig. 4). In these analyses, we used controls to avoid the influence of pG4s' lengths, which were randomly selected in non-pG4 promoters with the same number of sequences to that of pG4s, and the same length to the average length of pG4s. When analyzing the G-tracts in pG4s, under the same constraint condition, we used the all G-tracts' sequences in non-pG4 promoters as controls.

The distribution of frequency for variants mapped to each promoter region was extracted from gnomAD variant call files directly. P values for the difference between the expected number of variants per sequenced allele across genomic regions were called using a Chi-squared test. Only variants that did not overlap with annotated coding regions of other genes were compared to ensure that promoters overlapped with other gene coding regions were excluded.

**Sequence context modeling of substitution probabilities**. The sequence context model was constructed according to the method developed by Aggarwala et al.[33], with minor modifications. The intergenic sequences were defined as the full set of genomic sequences that were not annotated in GENCODE annotations (v38). We then removed telomeres (20 kb at the 5 'and 3' ends), repetitive regions and sequences that do not present in the accessibility mask (version 20160622) filter of the 1000 Genomes Project. Variants in the European subpopulation of 1000 Genomes project (EUR) were identified for use in downstream analysis.

Next, the statistical framework to model substitution probabilities as previously published[33] based on the heptamer sequence context was used to calculate posterior substitution probabilities for intergenic noncoding regions. Cumulative substitution probabilities for each mutation site in a heptamer context were

calculated by summing over all substitution probabilities for a given heptamer context, such as the sequence ATCGTCC showing the cumulative substitution probabilities of ATCCTCC, ATCATCC and ATCTTCC. The expected substitution probabilities in the regions, such as pG4s, were calculated by summing over the cumulative substitution probabilities of the heptamer contexts at each site. The ratio of observed versus expected substitution sites was calculated by the numbers of variants divided by the expected substitution probabilities in each region.

To get the values in the control zone, we first randomly selected the sequences in non-pG4 promoters using the same number and average length of pG4s, and determined their polymorphic sites. Then, we repeated this random selection and polymorphic site evaluation cycle for 10,000 times. For the obtained data, we removed the top and bottom 2.5% values, and the rest was used to build the control zone as the bootstrapped 95% confidence interval for the ratio.

**Population genomic analyses**. The HKA test was used to analyze whether a pG4 region in a 1 kb promoter or 0.2 kb promoter conforms to neutral evolution. The HKA test was slightly modified by referring to the method developed by Guiblet, et al.[21]. Specifically, we used fixed single-nucleotide variants between human and orangutan, and SNPs from high coverage genomic sequence data (>30× sequencing depth) for the 2504 unrelated samples in the 1000 Genomes Project[61]. A total of 95,738,090 fixed single-nucleotide variants between human and orangutan genomes were retrieved from the vertebrate MULTIZ 100-way alignment[62,63]. The total observed allele number was at least 80% of the maximum number of sequenced alleles and considered as plausible alleles. We discarded singletons and doubletons in the plausible alleles as they might possibly represent false positives. Fixed and polymorphic sites were intersected with 1 kb promoters and 0.2 kb promoters, respectively. Non-pG4s regions in all 1 kb and 0.2 kb promoters regardless pG4-containing statuses were used as background variations. We compared the observed counts of fixed and polymorphic sites at pG4 sequences to those expected values based on the non-pG4 regions in 1 kb and 0.2 kb promoters. Fisher's exact test was used to evaluate the significance of an odds ratio of fixed variants to polymorphic sites, as well as the 95% confidence interval of odds ratios.

A phylogenetic tree was constructed based on the information of 12 selected primates in the vertebrate MULTIZ 100-way alignment results from the UCSC database[62,64], and presented as a hierarchical clustering dendrogram when plotted by iTOL[65] (https://itol.embl.de/). We individually analyzed the fixed single-nucleotide variants between the human genome and each of these 11 primate genomes by genome-wide alignments, and obtained SNPs from the 2504 unrelated samples in the 1000 Genomes Project to perform HKA test.

Tajima's D, Fu and Li's D and Fu and Li's F tests were performed using the R package PopGenome[66] based on the SNPs from the 2504 unrelated samples in the 1000 Genomes Project[61]. We used non-pG4 regions in all promoters (including both pG4 and non-pG4 promoters) as controls.

**Positional constraint analyses**. For each individual nucleotide position in a G-tract consisting of three Gs in a pG4 sequence, mutability-adjusted proportion of singletons (MAPS) metric[41] was applied for its positional constraint analysis. Our model was constructed based on a reported MAPS model (https://bitbucket.org/biociphers/g4-paper-2019/src/master) with slight modifications by referring to a method developed by Lee, et al.[17]. Our MAPS model was also trained by regressing the percentage of singleton trinucleotide-related synonymous variants residing in protein-coding sequences of the human genome over the mutation rate of all types of trinucleotide sequences. Mutagenesis vulnerability of methylated cytosine was also evaluated for CpG dinucleotides. The filtering criteria of Variant frequency analysis was used to filter all variants for the positional constraint analyses.

For G-tracts with more than three Gs, the location of disrupted sites for each G-tract can be different and create complex situations. Therefore, we only considered the G-tracts composed of three Gs in our analysis. MAPS values were also determined for the set of variants with VEP predicted missense, or with predicted loss-of-function (pLoF) in genomAD. pLoF variants were defined as nonsense, frameshift, and canonical splice site shift (intronic +1, +2, −1, −2) variants. We calculated the MAPS for eight categories of pG4 variants: (1) all genes, (2) constitutive genes in which each transcript's promoter includes at least one pG4, (3) alternative genes that contain both pG4 and non-pG4 promoters, (4) stable-pG4s, (5) unstable-pG4s.

**Calculation of isoform expression of constitutive and alternative genes across tissues in GTEx**. A gene with each promoter containing at least one pG4 was defined as a constitutive gene, and a gene that has both pG4-containing promoter(s) and non-pG4 promoter(s) was defined as an alternative gene. The median expression of each annotated isoform (as measured in units of TPMs) across tissues was obtained from GTEx v8[45]. Median TPMs for each isoform were extracted for all-pG4- or non-pG4 containing 1 kb promoters and 0.2 kb promoters of each alternative gene, and the promoters driving the highest pG4- or non-pG4 isoform expression were selected. An isoform was considered expressed if its TPM in the tissue was greater than 1. The ratio of the isoform driven by pG4 1 kb or 0.2 kb promoter versus all expressed isoforms of an alternative gene in each tissue was then calculated.

**TT-seq, ChIP-seq, Cleavage Under Targets and Tagmentation (CUT&Tag) and RNA-seq analyses**. A published TMPyp4-treated CUT&Tag dataset and a TT-seq dataset (GSE178668[30]) were used to explore the relationship between G4 structures and gene expression. TT-seq pipeline was referred to the method developed by Schwalb, et al.[67] with slight modifications. Reads were demultiplexed and mapped with STAR 2.7.10a[68] to the hg38 genome assembly. Samtools[69] was used to qualify SAM files, where by alignments MAPQs smaller than 7 (-q 7) were skipped and only proper pairs (-f 99, -f 147, -f 83, -f 163) were selected. Next, the read counts across each gene were quantified by HTSeq 2.0[70] with the union mode, and DESeq2 was used to perform the differential gene expression analysis[71]. To estimate the fold changes based on the *Drosophila* spike-in RNA, size factors were calculated on the counts of the *Drosophila* genes and applied to the human gene counts prior to fold change estimation with DESeq2.

ChIP-seq and CUT&Tag reads were aligned to the human genome (UCSC hg38) with Bowtie2 version 2.4.2. Samtools was used to qualify SAM files, whereby the alignments with MAPQ smaller than 10 (-q 10) and secondary and partial alignments (-F 2308) were skipped. Next, replicate reads were marked and removed using SAMBAMBA version 0.6.6[72]. Peaks were called using MACS2 (model-based analysis of ChIP-Seq) version 2.2.6 with default parameters and a q-value cutoff of $1 \times 10^{-5}$. For the ChIP-seq data in K562 and HepG2 cells (GSE145090), we selected the overlapping peaks that were 5 out of 8 and 6 out of 9 replicated samples in K562 and HepG2 cells, respectively, using bedtools with version v2.30.0[73]. For CUT&Tag and other ChIP-seq data (GSE133379), the overlapping peaks were also selected in two replicate samples using bedtools. The all G4 peaks were annotated with bedtools intersect command. The different peaks analysis was performed using R package DiffBind and DESeq2. The genes with differential expression and genes with different G4 occupancy were identified by FDR < 0.05, and 252 genes were shared in those two groups of genes. GO enrichment analysis was performed using the clusterProfiler (v4.2.2) package[74].

RNA-seq reads were filtered using fastp software[75] (v0.23.2) with its default parameters, and transcript levels were quantified using the Salmon algorithm (v1.10.0). We defined the summation of transcript count values of the same promoter as the promoter activity, and then used DESeq2 to perform differential promoter activity analyses.

**cis-eQTLs enrichment**. We obtained the variant-gene pairs from the GTEx release version 8 (URL: https://gtexportal.org/home/datasets) that collected nominally significant *cis*-eQTLs. Among them, the Lead *cis*-eQTLs variants of each gene were determined as those with the lowest *P* values for that gene in each individual tissue, while the Causal *cis*-eQTLs variants were identified as those showing causal relationship with gene expression when screened by CaVEMaN, a computational model[48]. We overlapped the sets of the Lead, Causal and nominally significant variants with promoter pG4 and non-pG4 regions of promoters. Next, we quantitatively compared the significant *cis*-eQTL variants per region with the tested non-significant SNPs residing in the same region to calculate the proportion of *cis*-eQTLs versus non-*cis*-eQTL SNPs. A two-sided Fisher exact test was employed to evaluate the enrichment statistics of the *cis*-eQTLs in pG4 G-tracts and pG4s versus non-pG4 regions in pG4-containing promoters.

The directional bias of normally *cis*-eQTLs between G-tracts in promoter pG4s and non-G-tracts variants was compared by binarizing effect size precomputed for each *cis*-eQTLs by GTEx and comparing the proportion of *cis*-eQTLs with either a positive effect, or negative effects on gene expression in each feature. Statistical significance was determined by a two Fisher exact test.

**Enrichment analyses of histone marks, chromatin remodeler and transcript factor (TF) binding sites**. Genomic binding sites for histone modification marks, chromatin remodelers and TFs aligned to GRCh38/hg38 Reference Genomes were downloaded from ENCODE[25]. For histone modifications and histone H2A variant H2AFZ, 13 datasets comprised of 12 unique marks in K562, and 12 datasets comprised of 11 unique marks in HepG2 (with 1 repeated H3K4me3 dataset for each cell line). For chromatin remodelers, 79 datasets comprised of 59 unique marks in K562 (with 20 repeated datasets), and 63 datasets comprised of 56 unique marks in HepG2 (with 7 repeated datasets). For TFs, 490 datasets were comprised of 338 unique marks in K562 (with 152 repeated datasets), and 578 datasets were comprised of 505 unique marks (with 73 repeated datasets). To maximize the robustness of our analysis, only peaks called with an irreproducible discovery rate ≥540 were used for the following enrichment analysis. Randomization (10,000 times) for endogenous G4s or control sites and statistical analysis was performed using GAT[76] in the workspace (either 1 kb or 0.2 kb promoter regions). For marks that had been mapped in multiple different experiments, their merged peaks were considered.

**Statistics and reproducibility**. Data were analyzed and statistical evaluation was performed using and Python (version 3.10.9) and R (version 4.2.2) languages. Significant differences were denoted by asterisks. Detailed statistical tests or descriptions are provided in the custom codes in Code availability.

**Reporting summary**. Further information on research design is available in the Nature Portfolio Reporting Summary linked to this article.

## Data availability

The data used in the analyses of this study were downloaded from NCBI Gene Expression Omnibus (GEO; https://www.ncbi.nlm.nih.gov/geo/). The analyzed datasets include: BG4 ChIP-seq in K562 cells (GSE107690); BG4 ChIP-seq in HepG2 cells (GSE145090); G4P ChIP-seq in A549, H1975 and HEK293T cells (GSE133379); TMPyp4-treated CUT&Tag and TT-seq dataset in HEK293T cells (GSE178668); SG4 ChIP-seq in U2OS and K562 cells[77] (GSE207567); RNA-seq in U2OS[78] (GSE173483), K562[79] (GSE228464), A549[80] (GSE197555) and H1975[81] cells (GSE193258). The ChIP-seq datasets of histone modification marks, chromatin remodelers and TFs, and the RNA-seq datasets of HepG2 and K562 were downloaded from ENCODE (URL: https://www.encodeproject.org/), and the dataset names are listed in Supplementary Data 2. Genetic variation data from The Genome Aggregation Database version 3.1 release are available from https://gnomad.broadinstitute.org/downloads#v3. Genetic variation data from the 1000 Genomes Project are available from the European Nucleotide Archive (https://www.ebi.ac.uk/ena/data/view/PRJEB31736). Gene expression and cis-eQTL mapping data from the Genotype Tissue Expression Project version 8 release are available from the GTEx Portal website (URL: https://gtexportal.org/home/datasets). The source data to generate the graphs in Figs. 1f–h, 2a, b, 2d–g, 3–4, 5d and 6 are provided in Supplementary Data 3.

## Code availability

All analysis scripts used to generate the primary results and figures are available at GitHub (https://github.com/GuangyaoL/Promoter-pG4-analysis) and Zenodo (https://doi.org/10.5281/zenodo.7992629).

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

## Acknowledgements

This work was supported by the National Natural Science Foundation of China (82273107, 81872293) to G.S. and (81802798) to D.L., and the Fundamental Research Funds for the Central Universities (2572022DQ06) to G.S. We thank Dr. Daniel B. Stovall from College of Arts and Sciences, Winthrop University, Rock Hill, SC 29733, the United States, for critical reading of the manuscript.

## Author contributions

G.L. and Gu.S. initiated the project, wrote the manuscript and finalized the figures. G.L. analyzed the datasets and create figure drafts. Go.S., Y.W. and W.W. contributed to data collection, provided conceptual suggestions and critically read the manuscript. J.S. and D.L. contributed to general plan of the research direction, data collection, and critical revision of the manuscript. All authors have read and agreed to the published version of the manuscript.

## Competing interests

The authors declare no competing interests.
