## [Peer Review File · Communications Biology]

Reviewers' comments:

Reviewer #1 (Remarks to the Author):

The manuscript by Li et al. utilizes numerous computational methods to investigate the selective pressure of putative G4s (pG4s) in promoters and their interaction with gene expression. Using several large-scale genomic and genetic data resources, the authors demonstrate that, in promoter pG4s, the allele frequencies of G-tracts are remarkably lower than those of connecting loops. They also show that the central Gs in G-tracts are evolved under higher selection pressure than the other two Gs. Moreover, promoter pG4s are associated with enhanced promoter activity and are enriched for cis-eQTLs, as well as overlapping with binding sites of many chromatin remodelers and transcription factors. Although they provide many results with secondary data analyses to support the significant function of promoter pG4s in gene expression, experimental verification is necessary to illustrate some of the conclusions derived from calculation.

Main concerns:

1. In Fig. 4c, the authors conclude that "the presence of G4 structures in the proximity upstream of TSS is unfavorable for the promoters to be used in driving transcription." However, Fig. 4b shows that the proportions of transcript isoforms driven by pG4-containing 0.2kb promoters in alternative genes are higher than those in 1kb promoters in different tissue environments. Additionally, in Fig. 6c, more TF binding sites are detected in pG4-containing 0.2kb promoters than in 1kb promoters. These conclusions are contradictory, and additional analysis and discussion are necessary.

2. In line 456-457, "Thus, our results support the hypothesis that G4 structures in both 1kb and 0.2kb promoters enhance promoter activity." The authors reach this conclusion by analyzing the effects of G4s on promoter activity by calculating the ratios of promoter activity in K562 versus HepG2. However, the chromatin backgrounds in these two cells are different, and the presence of G4s is not the only factor for gene transcription. Therefore, it is insufficient to draw the conclusion that G4 structures enhance promoter activity by comparing the promoter activity in two cell lines. More evidence of the effect of G4 on promoter activity in the same system is needed. For example, in a certain cell line, the authors could compare the activities of certain G4-containing promoters to their corresponding sequences mutated in one or few bases by CRISPR so that they cannot form G4s anymore.

3. In line 530-531, "while variations in unstable-pG4s, cpG4s, and non-pG4s showed an insignificant effect." In Fig. 6d, the degree of decrease of cpG4s and unstable-pG4s to 1kb promoter is similar to all-pG4s and stable-pG4s to 1kb promoter, which is considered a significant change. Please mark the calculated significant differences.

Minor comments:

1. Line 461-495 - The description about TMPyP4-treated G4-CUT&Tag is confused. The authors mentioned "TMPyP4-treated CUT&Tag and TT-seq dataset" in Line 715, but they confused CUT&Tag and ChIP, which are two different methods.

1) Line 461 - "combined the BG4-ChIP and CUT&Tag" → "combined the BG4 antibody and CUT&Tag".

2) Line 466 - "the BG4 ChIP-seq and TT-seq datasets" → "the G4-CUT&Tag and TT-seq datasets".

3) Line 491-492 - "G4 signals obtained by ChIP-seq study using the G4-CUT&Tag approach" → "G4 signals obtained by G4-CUT&Tag approach".

4) Line 493-494 - "annotated genes enriched by ChIP-seq identified G4 signals" → "annotated genes enriched by G4-CUT&Tag identified G4 signals".

2. Line 612 - "Fig. 8, 9" → "Fig. 8 and Supplementary Fig. 9".

Reviewer #2 (Remarks to the Author):

The authors provide a well-written and novel work on selective constraints on G4s. They perform

thorough analyses of human promoters, eQTLs, epigenetic markers and TFBSs. In addition, they perform a large scale analysis of population variants, both rare and common. The manuscript is very interesting. However, there are certain limitations, most importantly putting their work in perspective and reflect the other works before them on the subject area, in which they do poorly. Also, the selection constraints have been shown before and it will be interesting to show an evolution aspect to their findings, e.g. by comparing primates.

Importantly, the authors also made an effort and provide a github repo for reproducibility purposes. However, this repo is not available limiting my ability to judge the quality of their work. They need to provide access to the github repo for reviewers.

Figure 1 does not provide any novelty. These graphs have been previously generated. I would suggest merging figures 1 and 2.

The authors need to provide a more complete introduction that summarizes other researchers' works on the subject. Some suggested articles below to discuss including relevance of G4s in selection constraints, eQTLs and sQTLs (especially PMID: 31988292, which is only briefly mentioned).

PMID: 31988292
PMID: 36604282
PMID: 35504902
PMID: 35573091
PMID: 34187812
PMID: 21455236

Fig5. volcano plots, the authors should highlight the top genes in the plots.

The part of the work in which the authors describe the association between G4s and TFBSs has previously been thoroughly shown. However, the epigenetic marker association is very interesting.

Can the authors provide a short examination of the evolutionary aspect of their work? How do these findings hold when comparing with another primate species?

Please provide more information on non-pG4 controls, how they were generated.

We would like to thank the reviewers for carefully reading and commenting on our manuscript. We provide our point-to-point response below to the reviewers' comments.

Reviewer #1

The manuscript by Li et al. utilizes numerous computational methods to investigate the selective pressure of putative G4s (pG4s) in promoters and their interaction with gene expression. Using several large-scale genomic and genetic data resources, the authors demonstrate that, in promoter pG4s, the allele frequencies of G-tracts are remarkably lower than those of connecting loops. They also show that the central Gs in G-tracts are evolved under higher selection pressure than the other two Gs. Moreover, promoter pG4s are associated with enhanced promoter activity and are enriched for *cis*-eQTLs, as well as overlapping with binding sites of many chromatin remodelers and transcription factors. Although they provide many results with secondary data analyses to support the significant function of promoter pG4s in gene expression, experimental verification is necessary to illustrate some of the conclusions derived from calculation.

Main concerns:

1. In Fig. 4c, the authors conclude that "the presence of G4 structures in the proximity upstream of TSS is unfavorable for the promoters to be used in driving transcription." However, Fig. 4b shows that the proportions of transcript isoforms driven by pG4-containing 0.2kb promoters in alternative genes are higher than those in 1kb promoters in different tissue environments. Additionally, in Fig. 6c, more TF binding sites are detected in pG4-containing 0.2kb promoters than in 1kb promoters. These conclusions are contradictory, and additional analysis and discussion are necessary.

Reply: We thank the reviewer for pointing this out for us. We apologize for our inappropriate interpretation of Fig. 4c (Lines 440-444 in the initially submitted manuscript).

(1) The analyses of Fig. 4c aimed to determine how pG4-containing 1kb and 0.2kb promoters are related to the differential (both increased and decreased) promoter activities of genes among different tumor tissues, or between tumor and normal tissues. Based on the OR values shown in Fig. 4c, we concluded that pG4-containing 1kb promoters contribute to "differential promoter activities" of these genes, while pG4-containing 0.2kb promoter likely maintain "invariable or constitutive activities among different promoters" of these genes. Thus, the conclusion of Fig. 4c was irrelevant to the preferential use of pG4 promoters in multi-promoter genes.

Again, we apologize for the inappropriate description of Fig. 4c in our initial

submission. In the revised manuscript, we provided the correct interpretation of Fig. 4c (Line 476-480) and also reorganized the Fig. 4c legend.

(2) Both Fig. 4b and Fig. 6c support the notion that pG4-containing promoters are preferentially used in multi-promoter genes.

1) The data in Fig. 4b showed that, in alternative genes, the isoform transcripts driven by pG4-containing 1kb and 0.2kb promoters occupied >72% and >77% of all transcripts, respectively.

2) The results in Fig. 6c showed that pG4-containing 1kb and 0.2kb promoters harbor significantly higher numbers of TF binding sites than the promoters without pG4.

2. In line 456-457, "Thus, our results support the hypothesis that G4 structures in both

1kb and 0.2kb promoters enhance promoter activity." The authors reach this conclusion by analyzing the effects of G4s on promoter activity by calculating the ratios of promoter activity in K562 versus HepG2. However, the chromatin backgrounds in these two cells are different, and the presence of G4s is not the only factor for gene transcription. Therefore, it is insufficient to draw the conclusion that G4 structures enhance promoter activity by comparing the promoter activity in two cell lines. More evidence of the effect of G4 on promoter activity in the same system is needed. For example, in a certain cell line, the authors could compare the activities of certain G4-containing promoters to their corresponding sequences mutated in one or few bases by CRISPR so that they cannot form G4s anymore.

Reply: We thank the reviewer for pointing this out for us. Certainly, functional validation of a subset of promoter G4s would be helpful to support the positive effects of G4s on promoter activity. However, it would be difficult to choose the number, types and locations of genes for promoter G4 mutagenesis studies, in order to make the selected genes or promoters representative. Meanwhile, the mutagenesis of these regulatory elements may introduce additional or unexpected chromatin alterations that affect promoter activities. Technically, G4 structures can impact CRISPR-Cas9 activity in terms of target recognition, R-loop progression and stability, and target dsDNA cleavage as reported in previous literature (Balci H, *et al.* Targeting G-quadruplex forming sequences with Cas9. *ACS Chem Biol.* 2021. PMID: 33769784). Therefore, these validation studies may not be completed during the revision period.

Nevertheless, we agree that the chromatin landscapes in K562 and HepG2 cell lines may have regional differences. Thus, to generalize our finding, we further analyzed the ChIP-seq and RNA-seq data of another two pairs of cell lines (U2OS versus K562, and A549 versus H1975), and obtained the results similar to that of K562 versus HepG2 analyses (shown in Supplementary Figure 8). Therefore, the analyses of these three pairs of cell lines strongly support that transcripts driven by G4-containing promoters exhibit significantly higher expression than those without G4s, and strongly suggest G4s' positive effects on promoter activity.

3. In line 530-531, "while variations in unstable-pG4s, cpG4s, and non-pG4s showed an insignificant effect." In Fig. 6d, the degree of decrease of cpG4s and unstable-pG4s to 1kb promoter is similar to all-pG4s and stable-pG4s to 1kb promoter, which is considered a significant change. Please mark the calculated significant differences

Reply: We thank the reviewer for the suggestion. For the bar graphs in Fig. 6d, we used the Fisher test to determine the OR values of gene expression changes caused by G-tract variations in different types of pG4s in promoters. Because each bar represents an individual datum, it is difficult to mark any significant difference without causing confusions or misinterpretation. Therefore, we added Supplementary Table 1 to provide the P values in Fig. 6d.

Minor comments:

1. Line 461-495 - The description about TMPyP4-treated G4-CUT&Tag is confused. The authors mentioned "TMPyP4-treated CUT&Tag and TT-seq dataset" in Line 715,

but they confused CUT&Tag and ChIP, which are two different methods.

Reply: We thank the reviewer for making these corrections. The errors have been corrected accordingly.

1) Line 461 - “combined the BG4-ChIP and CUT&Tag” → “combined the BG4 antibody and CUT&Tag”.

Reply: The error has been corrected (in Line 501-502 of the revised manuscript).

2) Line 466 - “the BG4 ChIP-seq and TT-seq datasets” → “the G4-CUT&Tag and TT-seq datasets”.

Reply: The error has been corrected (in Line 506 of the revised manuscript).

3) Line 491-492 - “G4 signals obtained by ChIP-seq study using the G4-CUT&Tag approach” → “G4 signals obtained by G4-CUT&Tag approach”.

Reply: The error has been corrected (in Line 532 of the revised manuscript).

4) Line 493-494 - “annotated genes enriched by ChIP-seq identified G4 signals” → “annotated genes enriched by G4-CUT&Tag identified G4 signals”.

Reply: The error has been corrected (in Line 534-535 of the revised manuscript).

2. Line 612 - “Fig. 8, 9” → “Fig. 8 and Supplementary Fig. 9”.

Reply: We thank the reviewer for pointing this out for us. The error has been corrected (in Line 654 of the revised manuscript).

Reviewer #2:

The authors provide a well-written and novel work on selective constraints on G4s. They perform thorough analyses of human promoters, eQTLs, epigenetic markers and TFBSs. In addition, they perform a large scale analysis of population variants, both rare and common. The manuscript is very interesting. However, there are certain limitations, most importantly putting their work in perspective and reflect the other works before them on the subject area, in which they do poorly. Also, the selection constraints have been shown before and it will be interesting to show an evolution aspect to their findings, e.g. by comparing primates.

1. Importantly, the authors also made an effort and provide a github repo for reproducibility purposes. However, this repo is not available limiting my ability to judge the quality of their work. They need to provide access to the github repo for reviewers.

Reply: We thank the reviewer for the positive comments and the question. Actually, we could still locate the “file” in the GitHub repository, but also realized that the file had a very large size, which could make it difficult to be retrieved and directly opened. Therefore, we have split the file into four relatively small files. After uploading them to the GitHub, we checked the files and found that they could be located and correctly opened.

2. Figure 1 does not provide any novelty. These graphs have been previously generated. I would suggest merging figures 1 and 2.

Reply: We thank the reviewer for the suggestion. In the revised manuscript, we have merged the previous Fig. 1 and Fig. 2 into new Fig. 1.

3. The authors need to provide a more complete introduction that summarizes other researchers' works on the subject. Some suggested articles below to discuss including relevance of G4s in selection constraints, eQTLs and sQTLs (especially PMID: 31988292, which is only briefly mentioned).

PMID: 31988292; PMID: 36604282; PMID: 35504902; PMID: 35573091;

PMID: 34187812; PMID: 21455236

Reply: We thank the reviewer for the suggestion. We have discussed the relevant contents in the studies and cited these papers in the Introduction section of the revised manuscript.

4. Fig5. volcano plots, the authors should highlight the top genes in the plots.

Reply: We thank the reviewer for the suggestion. We have marked the top 10 genes in the volcano maps of Fig. 5.

5. The part of the work in which the authors describe the association between G4s and TFBSs has previously been thoroughly shown. However, the epigenetic marker association is very interesting.

Can the authors provide a short examination of the evolutionary aspect of their work?

Reply: We thank the reviewer for the comments, and also thank the reviewer for suggesting the discussion of the revolutionary aspects. Actually, in the first paragraph of the Discussion section, we discussed the revolutionary pressure of promoter pG4s, which is summarized below:

(1) Both pG4s and G-tracts, especially the latter ones, in pG4-containing promoters were under negative selection or balance selection when evaluated by the Hudson-Kreitman-Aquade (HKA) test (shown in Fig. 2).

(2) The Tajima's D test, Fu and Li's D, and Fu and Li's F tests can be used to analyze whether a selected region is under selection pressure. Importantly, these tests can distinguish directional selection from balancing selection.

Using the Tajima's D test, Fu and Li's D, and Fu and Li's F tests, we identified that promoter pG4s and G-tracts were subject to directional selection (Table 1).

(3) When comparing the site frequency spectra of pG4 sequences and G-tracts versus non-pG4 sequences, we observed relatively high prevalence of polymorphic variants at low MAFs for pG4s, G-tracts and loops in both 1kb and 0.2kb promoters (Fig. 1g

and Supplementary Fig. 5). These results suggested pG4s and G-tracts in the promoters evolved under negative selection.

(4) When evaluating the selective pressure using a mutability-adjusted proportion of singletons (MAPS), we observed that the central guanines (Gs) in 3G G-tracts had higher selection pressure than the other two Gs (Fig. 3).

The overall examination is illustrated below:

6. How do these findings hold when comparing with another primate species?

Reply: We thank the reviewer for the constructive suggestion. In the UCSC database, the genomic data from 11 primate species are available and their hierarchical genetic relationship is displayed in Fig. 2c of the revised manuscript (see below).

We individually analyzed the fixed single-nucleotide variants between the human genome and each of these 11 primate genomes by genome-wide alignments. As shown in Fig. 2d and 2e, the odds ratios (ORs) for pG4s in 1kb promoters were greater than 1 in most of the primate species, while, for pG4s in 0.2kb promoters, most ORs were very close to 1. Importantly, the ORs for pG4 G-tracts in 1kb promoter were greater than 1 in all analyzed primate species, and pG4 G-tracts in 0.2kb promoters also showed ORs > 1 in 9 of these 11 species (Fig. 2f and 2g).

Thus, both pG4s and their G-tracts are under heightened selective pressure, and

pG4 G-tracts are more powerful elements than entire G4 sequences in the HKA test to evaluate selection pressure on pG4s.

7. Please provide more information on non-pG4 controls, how they were generated.

Reply: We thank the reviewer for the suggestion that can improve the clarity of this manuscript.

In our analyses, we used either the entire non-pG4 regions or randomly selected non-pG4 regions as controls in different scenarios, which is described below. We have integrated the description of the control selection and generation into the corresponding sections in *Methods*.

(1) In Figure 2:

(a) when analyzing the allele frequencies, we used the upper 90% bound of the observed versus expected (LOEUF) metric as the constraint for the subset of reference transcripts with non-pG4 G-tracts (Supplementary Figure 4). Additionally, to avoid the interference/influence of pG4s' lengths, in non-pG4 promoters, we randomly selected the same number of sequences to that of pG4s, and the length of these sequences was the same to the average length of pG4s. The information has been added in Line 820-825 of the revised manuscript.

(b) when analyzing the G-tracts in pG4s, under the same constraint condition, we used the all G-tracts' sequences in non-pG4 promoters as controls.

(c) In Fig. 2c, when analyzing the ratios of the observed versus expected polymorphic sites in different pG4s and their partitioned/related regions, we built a control zone for comparison. To get the values in this control zone, we first randomly selected the sequences in non-pG4 promoters using the same number and average length of pG4s, and determined their polymorphic sites. Then, we repeated this random selection and polymorphic site evaluation cycle for 10,000 times. For the obtained data, we removed the top and bottom 2.5% values, and the rest was used to build the control zone in Fig. 2c, designated as “the bootstrapped 95% confidence interval for the ratio”. The information has been added in Line 855-860 of the revised manuscript.

(2) For the HKA analyses in previous Fig. 3a and Fig. 3b (new Fig. 2a and Fig. 2b), and the Tajima’s D, Fu and Li’s D and Fu and Li’s F analyses in Table 1, we used non-pG4 regions in all promoters (including both pG4-containing and non-pG4 promoters) as controls. The information has been added in Line 874-875.

(3) For the *cis*-eQTLs enrichment analyses in previous Fig. 6a and Fig. 6b (now figure numbers remain the same), we used non-pG4 regions in pG4-containing promoters as controls. The information has been added in Line 977-979.

(4) When analyzing the enrichment of different histone modification markers, chromatin remodeling factors and transcription factors in the promoter G4 regions, we employed the GAT software, which requires the input for work space, to-be-evaluated enrichment regions and binding region of the abovementioned factors. With promoter regions as workspace, the GAT software could randomly select regions with the same length to the assessed enrichment region (G4) and evaluate the enrichment of the binding regions of the abovementioned factors in these selected regions. The selection and evaluation will be repeated 10,000 times to obtain the values for the binding or enrichment in the control regions. The ratios between observed binding regions versus sequence lengths in the enrichment regions of the controls were used as the final enrichment folds. The information has been added in Line 999-1001.

REVIEWERS' COMMENTS:

Reviewer #1 (Remarks to the Author):

The authors have addressed my concerns and I am supportive for its publication.

Reviewer #2 (Remarks to the Author):

Authors have addressed the comments.

Rebuttal

REVIEWERS' COMMENTS:

Reviewer #1 (Remarks to the Author):

The authors have addressed my concerns and I am supportive for its publication.

Reviewer #2 (Remarks to the Author):

Authors have addressed the comments.

Reply: We thank the reviewers for the positive remarks to our manuscript. We are impressed by the professional comments from the reviewers and the editors.